# Rapid and specific degradation of endogenous proteins in mouse models using auxin-inducible degrons

Lewis Macdonald[1†], Gillian C Taylor[1†], Jennifer Margaret Brisbane[1], Ersi Christodoulou[1], Lucy Scott[1], Alex von Kriegsheim[2], Janet Rossant[3], Bin Gu[4,5,6*], Andrew J Wood[1*]

[1]MRC Human Genetics Unit, Institute of Genetics and Cancer, University of Edinburgh, Edinburgh, United Kingdom; [2]Cancer Research UK Edinburgh Centre, Institute of Genetics and Cancer, University of Edinburgh, Edinburgh, United Kingdom; [3]Program in Developmental and Stem Cell Biology, Hospital for Sick Children, Toronto, Canada; [4]Department of Obstetrics, Gynecology and Reproductive Biology, Michigan State University, East Lansing, United States; [5]Department of Biomedical Engineering; Michigan State University, East Lansing, United States; [6]Institute for Quantitative Health Science and Engineering, Michigan State University, East Lansing, United States

*For correspondence:
gubin1@msu.edu (BG);
andrew.wood@igmm.ed.ac.uk
(AJW)

[†]These authors contributed
equally to this work

Competing interest: See page
21

Reviewing Editor: Guillaume
Pavlovic, PHENOMIN, Institut
Clinique de la Souris (ICS),
CELPHEDIA, France

**Abstract** Auxin-inducible degrons are a chemical genetic tool for targeted protein degradation and are widely used to study protein function in cultured mammalian cells. Here, we develop CRISPR-engineered mouse lines that enable rapid and highly specific degradation of tagged endogenous proteins in vivo. Most but not all cell types are competent for degradation. By combining ligand titrations with genetic crosses to generate animals with different allelic combinations, we show that degradation kinetics depend upon the dose of the tagged protein, ligand, and the E3 ligase substrate receptor TIR1. Rapid degradation of condensin I and II – two essential regulators of mitotic chromosome structure – revealed that both complexes are individually required for cell division in precursor lymphocytes, but not in their differentiated peripheral lymphocyte derivatives. This generalisable approach provides unprecedented temporal control over the dose of endogenous proteins in mouse models, with implications for studying essential biological pathways and modelling drug activity in mammalian tissues.

## Editor's evaluation

This manuscript will be of interest to the broad class of biologists and especially mouse geneticists who study the function of protein-coding genes. The authors confirm the utility of the auxin-inducible degron tool to rapidly degrade the target protein of interest by developing genetically modified mouse models. This expands the set of tools to study gene function in a cell/tissue type, in adults (bypassing embryonic lethality) and also to more finely dissect the different functions of pleiotropic genes.

## Introduction

Methods to conditionally control gene function are an important part of the genetic toolbox in a wide range of experimental model systems. In rodents, conditional approaches typically make use of recombinases such as CRE and FLP, which allow the controlled excision, inversion, or translocation

of DNA flanked by recombinase target sites (**Kos, 2004**). Despite the immense contribution that recombinase-based systems have made to mouse genetics, their utility in studies of protein function is fundamentally limited by the fact that they target DNA.

DNA manipulations impact protein function with kinetics that are determined by the natural half-life of pre-existing mRNA and protein molecules. Chemical genetic approaches such as the auxin-inducible degron (AID) provide a potential solution to these problems (**Natsume and Kanemaki, 2017**; **Nishimura et al., 2009**). Auxins (e.g. indole-3-acetic acid [IAA]) are plant hormones that bind to TIR1, the substrate receptor subunit of a Cullin Ring E3 ubiquitin ligase complex (**Tan et al., 2007**). IAA binding greatly increases the affinity of TIR1 for target proteins containing a degron polypeptide (**Tan et al., 2007**), leading to the formation of a ternary complex, ubiquitination of the target protein, followed by its rapid degradation by the proteosome.

Pioneering work by the Kanemaki laboratory showed that this plant-specific system can be co-opted to conditionally degrade target proteins in non-plant species (**Natsume et al., 2016**; **Nishimura et al., 2009**). This is achieved by genetically fusing short degron tags (44 amino acids **Morawska and Ulrich, 2013**; **Nora et al., 2017**) to a protein of interest in cells that heterologously express a plant *Tir1* transgene. Addition of IAA ligand then induces rapid degradation of the degron-tagged protein, often with a half-life of less than 30 min, in a manner which is both reversible and dosage controllable (**Nishimura et al., 2009**). Other degron tag systems such as dTAG (**Nabet et al., 2020**; **Nabet et al., 2018**), and HaloPROTAC (**Buckley et al., 2015**) work on a similar conceptual basis, albeit via different molecular mechanisms.

These genetically encoded strategies for targeted protein degradation have revolutionised functional studies of essential proteins in cultured mammalian cells and invertebrates, providing insights into a range of 'fast' processes such as transcription (**Boija et al., 2018**; **Narita et al., 2021**), chromosome looping (**Gibcus et al., 2018**; **Nora et al., 2017**), and the cell cycle (**Hégarat et al., 2020**). However, targeted protein degradation in genetically engineered mammals remains in its infancy (**Gu et al., 2018**; **Yesbolatova et al., 2020**), and it has not yet been possible to degrade tagged endogenous proteins in adult tissues. Such an ability would enable protein function to be compared across a wide range of 'normal' cell types and disease models.

In a proof-of-principle study, we (BG and JR) previously observed degradation of transcription factors in early mouse embryos that were mosaic for expression of both TIR1 and the AID-tagged target protein (**Gu et al., 2018**). More recently, a modified AID system (AID2) was shown to induce degradation of a randomly integrated *GFP* transgene in adult and embryonic mouse tissues following intraperitoneal (I.P.) ligand injection (**Yesbolatova et al., 2020**). However, the original and more extensively characterised AID system was not tested in this study due to difficulties in deriving stable mouse lines expressing *Oryza sativa* TIR1.

Here, we derive novel transgenic mouse lines to show that the original AID system is highly effective for acutely depleting endogenously expressed proteins in adult tissues, embryos, and primary cells. Using a proteome-wide approach, we show that AID is highly specific for the target protein in vivo. Mechanistically, we find that the dosage of TIR1, IAA ligand and the AID-tagged target protein are all key determinants of degradation efficiency in primary cells. We then focus on the two mammalian condensin complexes to show that IAA-responsive mice allow the comparison of 'essential' protein function over short time-scales across cell lineages, and at different stages of differentiation.

## Results
### Generation of mouse models for auxin-inducible degradation of condensin subunits

The condensin I and II complexes (*Figure 1A*) are essential for mitotic chromosome formation and chromosome segregation in vertebrate cells (**Ono et al., 2003**), and are thought to work via a DNA-dependent motor activity to generate loops in chromosomal DNA (**Gibcus et al., 2018**; **Terakawa et al., 2017**). AID tagging of condensin subunits has enabled the consequences of their acute depletion to be studied in various cancer cell lines (**Gibcus et al., 2018**; **Samejima et al., 2018**; **Takagi et al., 2018**). However, it has been challenging to compare the functional requirement for condensins, or indeed other essential proteins, during cell division in different somatic cell lineages.

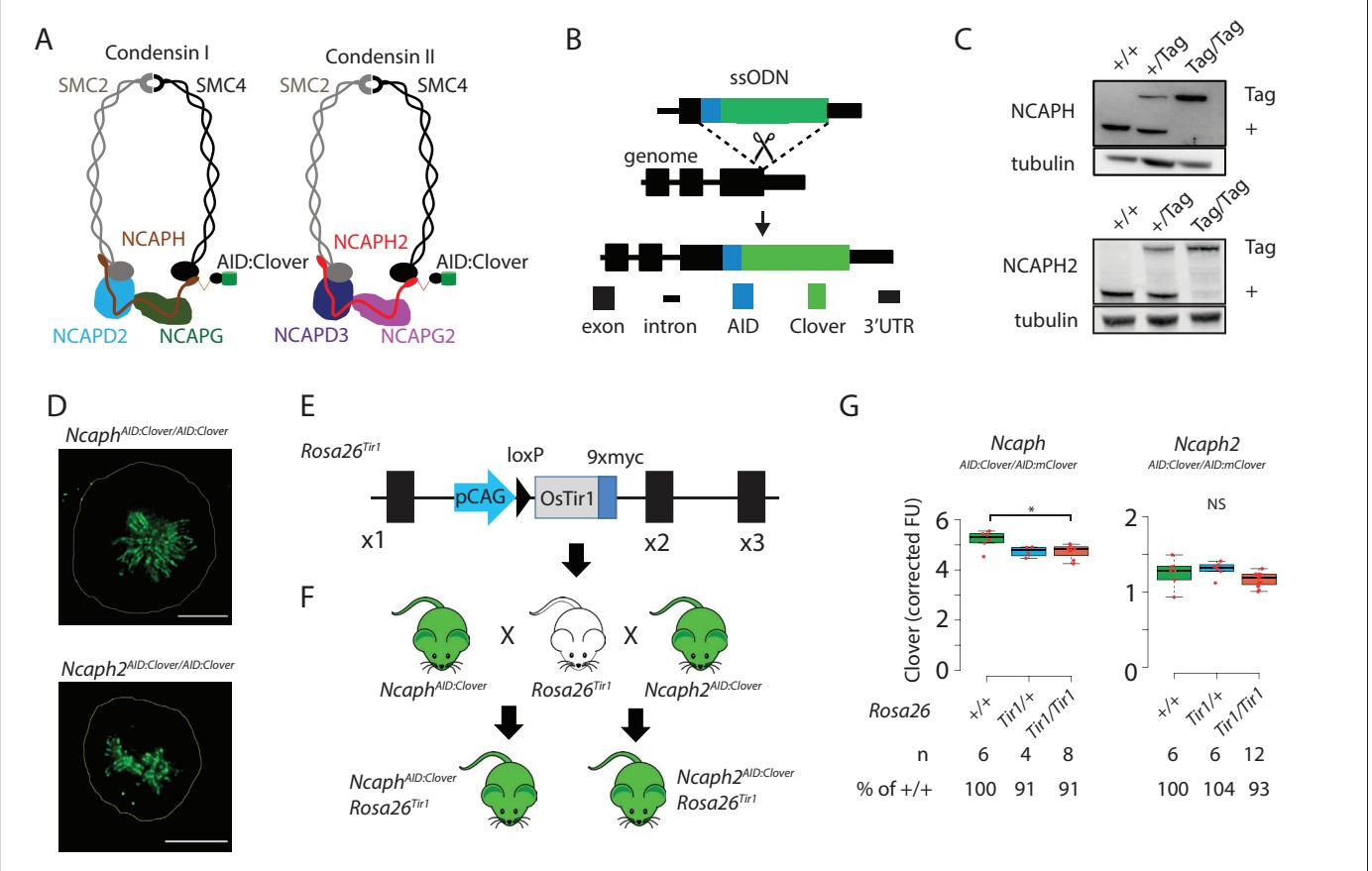

**Figure 1.** Mouse models for auxin-inducible degradation of condensin proteins. (**A**) Schematic diagrams showing the subunit composition of condensin I and II complexes with C-terminal AID:Clover. The kleisin subunits of condensin I and II are NCAPH and NCAPH2, respectively. (**B**) CRISPR-Cas9 strategy for integrating mClover cassettes at the *Ncaph* and *Ncaph2* loci using long single stranded deoxyoligonucleotides (ssODN) to generate *Ncaph*^AID:Clover and *Ncaph2*^AID:Clover alleles. Full details and sequences for the integrated cassettes are given in ***Supplementary file 3***. (**C**) Western blots prepared from thymic whole cell protein extract were probed with antibodies recognising endogenous NCAPH or NCAPH2, with tubulin as a loading control. '+' indicates wildtype allele, 'tag' indicates AID:Clover. (**D**) Immunofluorescence imaging of mitotic murine embryonic fibroblast lines derived from *Ncaph*^AID:Clover/AID:Clover and *Ncaph2*^AID:Clover/AID:Clover embryos. Scale bar = 5 μm. (**E**) Schematic diagram showing the *Rosa26*^Tir1 allele. Details on how this allele was generated are detailed in ***Figure 1—figure supplement 1D*** and Materials and methods. (**F**) Breeding scheme to combine endogenously tagged *Ncaph* and *Ncaph2* alleles with *Rosa26*^Tir1. (**G**) Clover fluorescence was measured by flow cytometry in primary S/G2/M thymocytes (gated on DNA content, *n* > 1000 cells/sample) from mice homozygous for AID:Clover-tagged target proteins, in combination with 0, 1, or 2 alleles of the Rosa26^Tir1 transgene. Cells were not subjected to IAA treatment. Boxplots show background-corrected mean fluorescence values from (*n*) biological replicate samples. * indicates a significant (p < 0.05) difference between genotypes (one-way analysis of variance [ANOVA] with Tukey HSD test, p < 0.05). NS: not significant.

The online version of this article includes the following figure supplement(s) for figure 1:

**Figure supplement 1.** Mouse models for auxin-inducible degradation of condensin proteins.

To address this, we generated mice in which the function of each condensin complex could be perturbed by IAA-mediated targeted proteolysis. The Easi-CRISPR approach (***Miura et al., 2018***; ***Quadros et al., 2017***) was used to generate two transgenic lines in which cassettes encoding the mini-auxin-inducible degron and Clover fluorescent protein (AID:Clover) were fused to the C-terminus of endogenous condensin subunits via a short flexible linker peptide (NCAPH^AID:Clover and NCAPH2^AID:-Clover, ***Figure 1A, B***, Supplemental Methods). The kleisin subunits NCAPH and NCAPH2 were selected for tagging as they are expressed at levels that are limiting for holocomplex assembly (***Walther et al., 2018***), and are known to be essential for condensin complex function in mice (***Houlard et al., 2015***; ***Nishide and Hirano, 2014***). Degradation of NCAPH and NCAPH2 should therefore ablate the function of condensin I and II, respectively.

In line with previous studies in cultured cells (*Gibcus et al., 2018*; *Samejima et al., 2018*; *Walther et al., 2018*), C-terminal tagging did not substantially affect steady-state expression (*Figure 1C*), or localisation to mitotic chromosomes (*Figure 1D*), for either fusion protein. Mice homozygous for the tagged alleles were born at the expected Mendelian frequency from crosses of heterozygous parents (*Figure 1—figure supplement 1A*), whereas null mutations in either target gene are known to cause embryonic lethality in the homozygous state (*Houlard et al., 2015*; *Nishide and Hirano, 2014*). *Ncaph^AID:Clover* homozygotes had similar litter sizes (*Figure 1—figure supplement 1B*), and were not growth impaired (*Figure 1—figure supplement 1C*) compared to heterozygotes, but *Ncaph2^AID:Clover* homozygotes were less fertile and smaller (*Figure 1—figure supplement 1B, C*). No other developmental abnormalities were observed in either line, indicating that the essential functions of NCAPH and NCAPH2 may be largely retained by the tagged proteins.

In parallel, we generated transgenic animals expressing the *Oryza sativa* (*Os*) *TIR1* gene constitutively by targeting a *TIR1* expression construct to the *Rosa26* locus (*Figure 1E*, *Figure 1—figure supplement 1D*) by microinjection of CRISPR-Cas9 reagents and a plasmid donor into mouse embryos at the two-cell stage (*Gu et al., 2018*; *Gu et al., 2020*). Initially, we attempted but failed to generate a mouse line constitutively expressing the *OsTIR1* gene at the *Rosa26* locus driven by the CAG promoter, consistent with other reports (*Yesbolatova et al., 2020*). We reasoned that the transient overexpression of TIR1 from the many copies of injected donor plasmids may have led to non-specific protein degradation in embryos and prevented us from generating live founder pups. We circumvented this hurdle by first generating a single-copy insertion of a conditionally activatable Lox-STOP-Lox (LSL) allele (*Rosa26-CAG-LSL-TIR1-9myc*, *Figure 1—figure supplement 1D*). The *Rosa26-CAG-LSL-TIR1-9myc* mice were then bred with a constitutive Cre line (*pCX-NLS-Cre*; *Belteki et al., 2005*) to remove the LSL cassette and generate the *R26-CAG-TIR1-9myc* mouse line (hereafter *Rosa26^Tir1*). Western blotting revealed broad expression of TIR1 across tissues (*Figure 1—figure supplement 1E*). However, although TIR1 expression is driven by a constitutive CAG promoter from the *Rosa26* safe harbor locus, we did observe differential expression levels among the tissues tested (*Figure 1—figure supplement 1E*), consistent with previous observations for LacZ reporters and creERT2 (*Hameyer et al., 2007*; *Mao et al., 1999*). As shown below, the expression of TIR1 in relevant tissues appears to be sufficient to facilitate target protein degradation. However, further characterisations are required to detail the expression pattern of TIR1 at tissue and single-cell levels to comprehensively characterise the TIR1 knock-in mouse lines and facilitate broader applications.

In order to generate IAA-responsive mice, we performed crosses to combine the *Rosa26^Tir1* allele with either *Ncaph^AID:Clover* or *Ncaph^AID:Clover* in double transgenic animals (*Figure 1F*). The presence of TIR1 has previously been reported to induce 'basal' degradation of some AID-tagged target proteins even in the absence of exogenous IAA (*Mendoza-Ochoa et al., 2019*; *Sathyan et al., 2019*; *Yesbolatova et al., 2020*). Importantly, the presence of *Rosa26^Tir1* alleles had little (NCAPH) or no (NCAPH2) effect on target protein expression (*Figure 1G*). Accordingly, double homozygotes were obtained at, or greater than, the expected Mendelian frequencies from crosses of *Ncaph-* or *Ncaph2^AID:Clover/AID:Clover*; *Rosa26^Tir1/+* parents (*Figure 1—figure supplement 1F*), and both sexes were fertile (*Figure 1—figure supplement 1G*). No significant difference in weight was observed between animals with 0, 1, or 2 alleles of *Rosa26^Tir1* (*Figure 1—figure supplement 1H*). Thus, despite low-level affinity of TIR1 for the degron peptide in the absence of auxin (*Tan et al., 2007*), we conclude that the *Rosa26^Tir1* transgene did not cause levels of auxin-independent degradation that were sufficient to induce overt phenotypes in combination with either of the AID-tagged target proteins studied here. However, further comprehensive phenotyping following the guidelines and protocols established by international programmes such as International Mouse Phenotyping Consortium (*Birling et al., 2021*), will be needed in the future to determine whether any of these genetic modifications induce phenotypic effects that were not detected in this study.

## Rapid degradation of AID-tagged endogenous proteins in primary cells

The ability of IAA to induce targeted protein degradation was then tested in short term cultures of primary CD8+ thymocytes, embryonic fibroblasts, and neural stem cells harvested from animals homozygous for *Rosa26^Tir1* and either AID-tagged allele (*Figure 2A*). In each case, addition of IAA to the culture media resulted in near complete (>90%) protein degradation within 2 hr (*Figure 2B–E*).

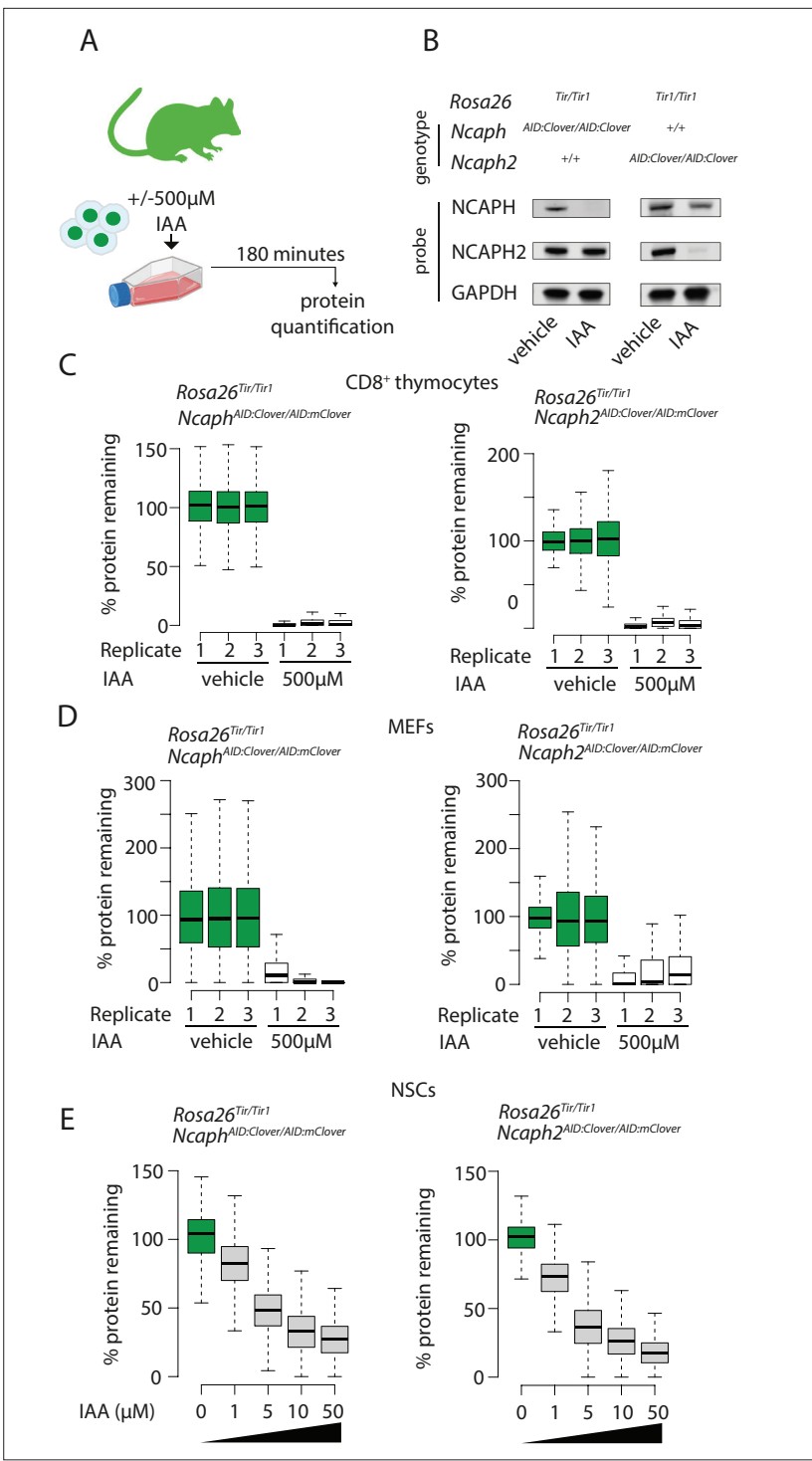

**Figure 2.** Rapid and titratable degradation of endogenous NCAPH and NCAPH2 in primary cells. (**A**) Schematic illustration of experiments designed to test targeted degradation of condensin subunits in primary cells. (**B**) Western blots prepared from thymus whole cell extract and probed with polyclonal antibodies against NCAPH, NCAPH2, or a GAPDH loading control. Robust tag-dependent degradation of target proteins is clearly evident after 3 hr of auxin treatment. (**C** and **D**) Boxplots quantify the extent of targeted protein depletion following IAA treatment (500 µM for 3 hr), measured by flow cytometry in primary CD8[+] thymocytes (**C**) and murine embryonic fibroblasts (MEFs - **D**). $n$ = 3 biological replicates from at least 2 independent experiments, with degradation measured in over 1000 S/G2/M cells in each case. To calculate % protein remaining, the background-corrected fluorescence value of each cell was expressed as a percentage of the mean fluorescence value for all cells in the

*Figure 2 continued on next page*

*Figure 2 continued*

vehicle-only condition. Boxes show the boundaries of upper and lower quartiles and whiskers show the range. Where negative values were observed (e.g. in MEFs due to variable autofluorescence between lines), a value of 0% was assigned. (**E**) Titration of target protein levels in primary neural stem cells treated with different IAA concentrations for 2 hr. Boxplots were generated as described for panels C and D.

## Dosage of ternary complex components determines degradation efficiency

AID and other degron tagging approaches achieve protein degradation through the formation of a ternary complex comprising a ligand, an E3 ligase substrate receptor, and the degron-tagged target protein (*Figure 3A*, *Tan et al., 2007*). Complex formation induces ubiquitination of target proteins via an E3 ligase: for the AID system this is SCF$^{Tir1}$. It is well established that the kinetics of protein degradation are determined, and can be experimentally manipulated by, ligand dose (*Natsume and Kanemaki, 2017*; *Nishimura et al., 2009*). We confirmed this finding in primary neural stem cells derived

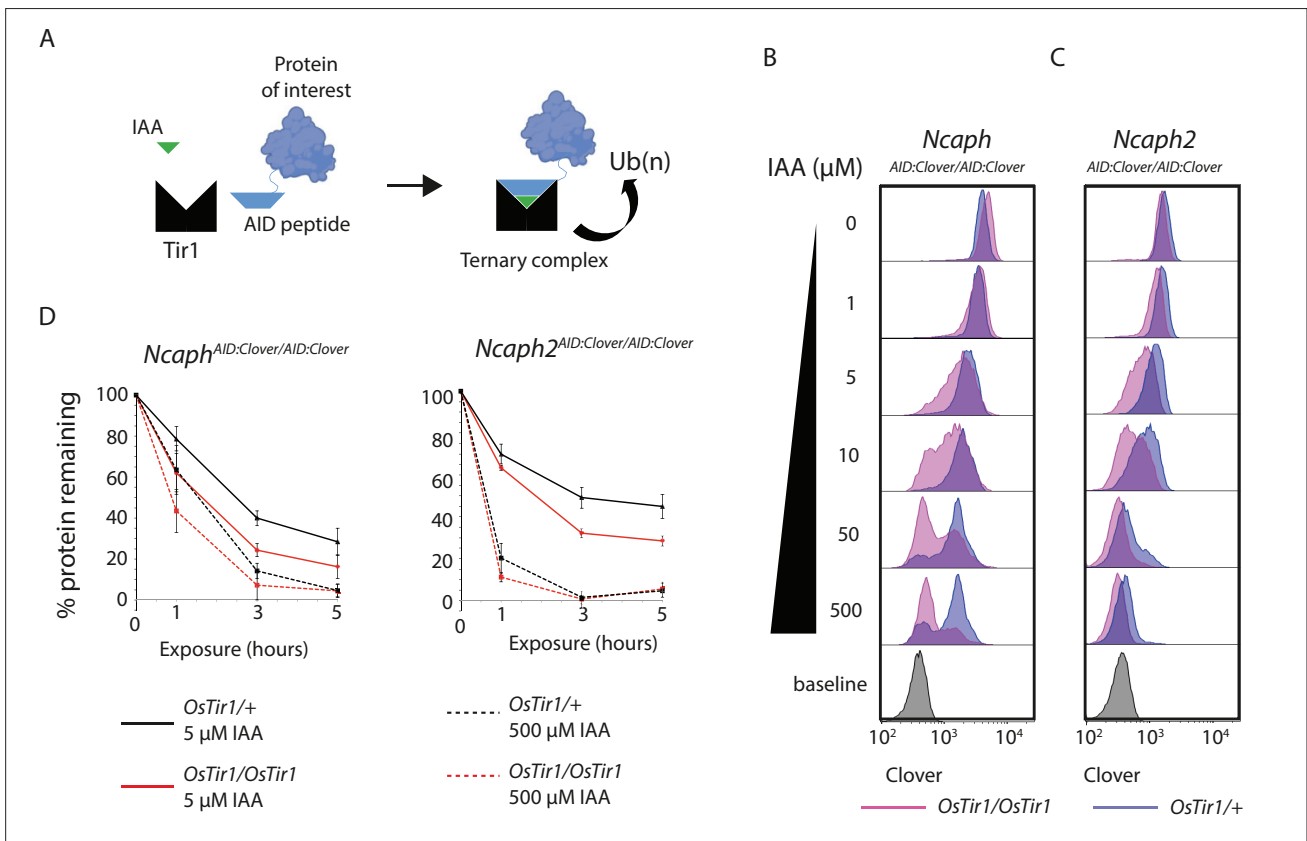

**Figure 3.** TIR1 dosage determines degradation kinetics of auxin-inducible degron (AID)-tagged proteins. (**A**) Schematic diagram illustrates the assembly of the Tir1 substrate receptor protein, IAA ligand and AID-tagged target protein-of-interest into a ternary complex necessary for target protein ubiquitination via SCF$^{Tir1}$, and degradation. (**B** and **C**) Histograms show the distribution of Clover expression levels, measured by flow cytometry in S/G2/M thymocytes cultured for 2 hr ex vivo in the presence of different IAA concentrations. Thymocytes were isolated from animals homozygous for either (**B**) *Ncaph*$^{AID:Clover}$ or (**C**) *Ncaph2*$^{AID:Clover}$ alleles in combination with either one (dark purple) or two (light purple) alleles of *Rosa26*$^{Tir1}$. Equivalent data from animals heterozygous for AID-tagged alleles are shown in *Figure 5—figure supplement 1*. (**D**) Comparison of depletion kinetics in the presence of one (black) versus two (red) alleles of the Tir1 transgene at low (solid line) versus high (dashed line) ligand concentrations (*n* = 3 biological replicate samples). Each experiment in panels B–D used data from at least 1000 S/G2/M thymocytes, gated on DNA content. In panel D, the mean background-corrected fluorescence value for each cell population is expressed as a percentage of the mean background-corrected fluorescence value for the vehicle only condition.

The online version of this article includes the following figure supplement(s) for figure 3:

**Figure supplement 1.** Histograms show the distribution of Clover expression levels, measured by flow cytometry in >1000 S/G2/M thymocytes cultured for 2 hr ex vivo in the presence of different IAA concentrations.

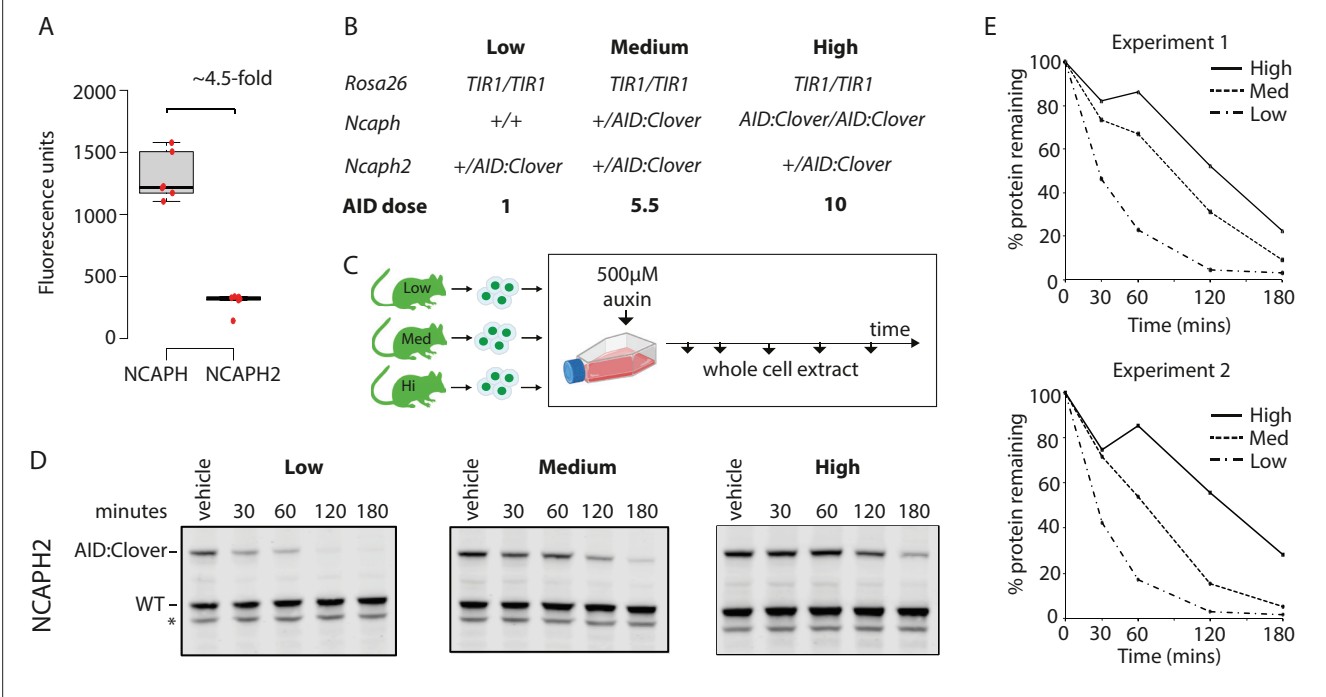

**Figure 4.** Dosage of auxin-inducible degron (AID)-tagged substrate proteins determines degradation kinetics. (**A**) The relative expression of NCAPH and NCAPH2 (n = 6 biological replicates each) in thymocytes, based on flow cytometric Clover fluorescence measurements in >1000 cells. (**B**) Table showing the relative total dose of AID-tagged proteins in mice heterozygous for $Ncaph2^{AID:Clover}$ in combination with either 0 (Low), 1 (Medium), or 2 (High) alleles of $Ncaph^{AID:Clover}$. Relative AID dose is calculated based on data in panel A. (**C**) Schematic showing the time course for auxin treatment of primary thymocytes in panels D and E. (**D**) Western blots probed with a polyclonal antibody against NCAPH2. Tagged protein (upper band) is degraded, whereas wildtype protein (lower band) is not. * indicates non-specific band. (**E**) Quantification of NCAPH2-AID:Clover depletion in the presence of low, medium, or high overall AID-tagged protein dose. Density of the AID:Clover band (see panel D) was first measured relative to the corresponding wildtype allele (bottom) as an internal control. The AID:WT ratio in the vehicle only control was set at 100% and IAA treatment conditions were then calculated relative to this value. Data from two independent experiments are presented.

from $Ncaph^{AID:Clover/AID:Clover}$ and $Ncaph2^{AID:Clover/AID:Clover}$ animals homozygous for $Rosa26^{Tir1}$ (**Figure 2E**). Whether the kinetics of AID-mediated protein degradation are also controlled by the cellular dosage of the substrate protein and/or the TIR1 substrate receptor is less well understood.

To address the role of substrate receptor dosage, we compared the efficiency of target protein degradation in primary thymocytes derived from animals homozygous for $Ncaph^{AID:Clover}$ or $Ncaph2^{AID:Clover}$ in combination with either 1 or 2 copies of the $Rosa26^{Tir1}$ transgene. This analysis revealed that two copies of $Tir1$ resulted in more efficient degradation of target proteins compared to a single copy (**Figure 3B–D**). The same trend was observed consistently for both NCAPH and NCAPH2 target proteins (**Figure 3B, C**), in the presence of either one or two AID-tagged alleles (**Figure 3B, C**, **Figure 3—figure supplement 1**). Depletion kinetics of AID-tagged mammalian proteins are therefore controlled not only by ligand dose, but also by dosage of the E3 ligase substrate receptor protein TIR1.

To determine the effect of target protein dosage on degradation efficiency, we designed a competition experiment which took advantage of the two distinct AID-tagged alleles that were available. The relative steady-state expression level of tagged NCAPH and NCAPH2 proteins was first quantified by flow cytometric measurement of Clover fluorescence in primary thymocytes. This revealed that the ratio of NCAPH to NCAPH2 is approximately 4.5 : 1 (**Figure 4A**), which is consistent with previous measurements in HeLa cells (**Walther et al., 2018**). Next, we performed crosses to generate $Ncaph2^{AID:mClover/+}$ animals carrying either 0, 1, or 2 alleles of $Ncaph^{AID:mClover}$ on a $Rosa26^{Tir1/Tir1}$ background, which allowed us to ask how the degradation kinetics of a constant dose of NCAPH2^{AID:Clover} protein are affected by the addition of NCAPH^{AID:Clover} proteins at increasingly high dose (**Figure 4B, C**). These experiments confirmed that NCAPH2 degradation was less efficient when the overall quantity of AID-tagged protein increased (**Figure 4D, E**). In summary, dosage of all three components of

the ternary complex can be limiting, and therefore control the degradation kinetics of AID-tagged target proteins.

## Comparing essential gene function between primary cell types

Loss of function mutations in condensin subunits cause fully penetrant embryonic lethality in mice (*Houlard et al., 2015*; *Nishide and Hirano, 2014*; *Smith et al., 2004*), but it is not known whether each complex is absolutely required for cell division throughout development. Our system for rapidly depleting essential condensin I and II subunits in different primary cell types enabled us to address this question. A BrdU pulse chase assay was established to assess the efficiency of cell division during a single-cell cycle across primary cell types.

We chose to focus on lymphocyte development, specifically comparing how rapid degradation of either a condensin I or a condensin II subunit affected cell division in precursor versus mature splenic cells in both the B- and T-cell lineages. Explanted cells from the bone marrow (precursor B), thymus (precursor T), or spleen (peripheral B and T) were cultured for 2 hr in the presence or absence of auxin to degrade either NCAPH or NCAPH2, then subjected to a 30-min BrdU pulse followed by washout and chase (*Figure 5A*). BrdU and DNA content were then measured by flow cytometry (*Figure 5B*). Over time, a fraction of BrdU⁺ cells divide to form G1 daughter cells with 2n DNA content (*Figure 5C*). If loss of either condensin complex inhibits cell division, IAA treatment should reduce the fraction of BrdU⁺ cells that progress through mitosis into G1 during the chase (*Figure 5D*). By quantifying and comparing the extent of this reduction across cell types (*Figure 5E*), we tested their ability to complete a single-cell division in the near absence of condensin I or II. As expected, acute degradation of either NCAPH or NCAPH2 did not affect BrdU incorporation (*Figure 5—figure supplement 1A*) or induce markers of DNA damage during S phase (*Figure 5—figure supplement 1B*), but instead caused an accumulation of 4N cells (*Figure 5D*) consistent with a cell cycle block during G2/M.

In primary cells from $Ncaph^{AID:Clover/AID:Clover}$; $Rosa26^{Tir1/Tir1}$ mice, acute depletion of an essential condensin I subunit impacted cell division to a greater extent in precursor T cells isolated from the thymus compared to activated mature T cells isolated from the spleen (*Figure 5D–F*). In thymocytes, where most cell division occurs at the 'beta-selection' stage of T-cell differentiation (*Kreslavsky et al., 2012*), treatment caused an 88% reduction (21.6% vs 3.1%, *Figure 5E, F*) in the fraction of G1 cells among the BrdU⁺ population following a 3.5-hr chase. In contrast IAA treatment of activated splenic T cells caused only a 22% reduction in this population (39% vs 31% of BrdU⁺ cells in G1, *Figure 5E, F*). By the same measure, precursor B cells isolated from the bone marrow were significantly more sensitive to NCAPH depletion compared to mature B cells isolated from the spleen (67% vs 48% reduction in BrdU⁺ G1 cells, respectively, *Figure 5E, F*). The same experiments repeated in primary cells from $Ncaph2^{AID:Clover/AID:Clover}$; $Rosa26^{Tir1/Tir1}$ animals revealed similar trends, with precursors more sensitive to condensin perturbation compared to mature cells, albeit with less profound effect sizes.

The observed differences between cell types were not attributable to differences in the extent of protein degradation, which were similar between precursor and peripheral cell populations (*Figure 5—figure supplement 1C*). However, despite the relatively mild consequences of condensin degradation on peripheral lymphocytes over a single-cell division (*Figure 5D–F*), cell trace experiments still showed a clear impact on proliferation over several cell cycles (*Figure 5—figure supplement 1D*). Altogether, these experiments show that lymphocytes at later stages of differentiation are better able to complete a single round of cell division in the near absence of either condensin I or II compared to their respective precursor cell populations.

## Acute degradation of AID-tagged proteins in living mice

Having established the utility of the AID system to compare essential protein functions between primary cell types, we next investigated its use in living adult mice. Because condensin expression is largely restricted to proliferating cells in adult tissues, we initially focused on haematopoietic organs where dividing cells are abundant. In a pilot dose-finding study, adult $Ncaph^{AID:Clover/+}$ $Rosa26^{Tir1/Tir1}$ mice received a single dose of IAA via I.P. injection, then thymus tissue was collected 2 hr later to quantify Clover fluorescence using flow cytometry. Increasing levels of protein degradation were observed as the dose was increased from 50 to 100 mg kg⁻¹ (*Figure 6—figure supplement 1A*). All subsequent experiments were therefore performed using the 100 mg kg⁻¹ dose. To evaluate hepatic toxicity of IAA, a panel of liver function tests was performed on plasma collected from animals either 2 or 72 hr

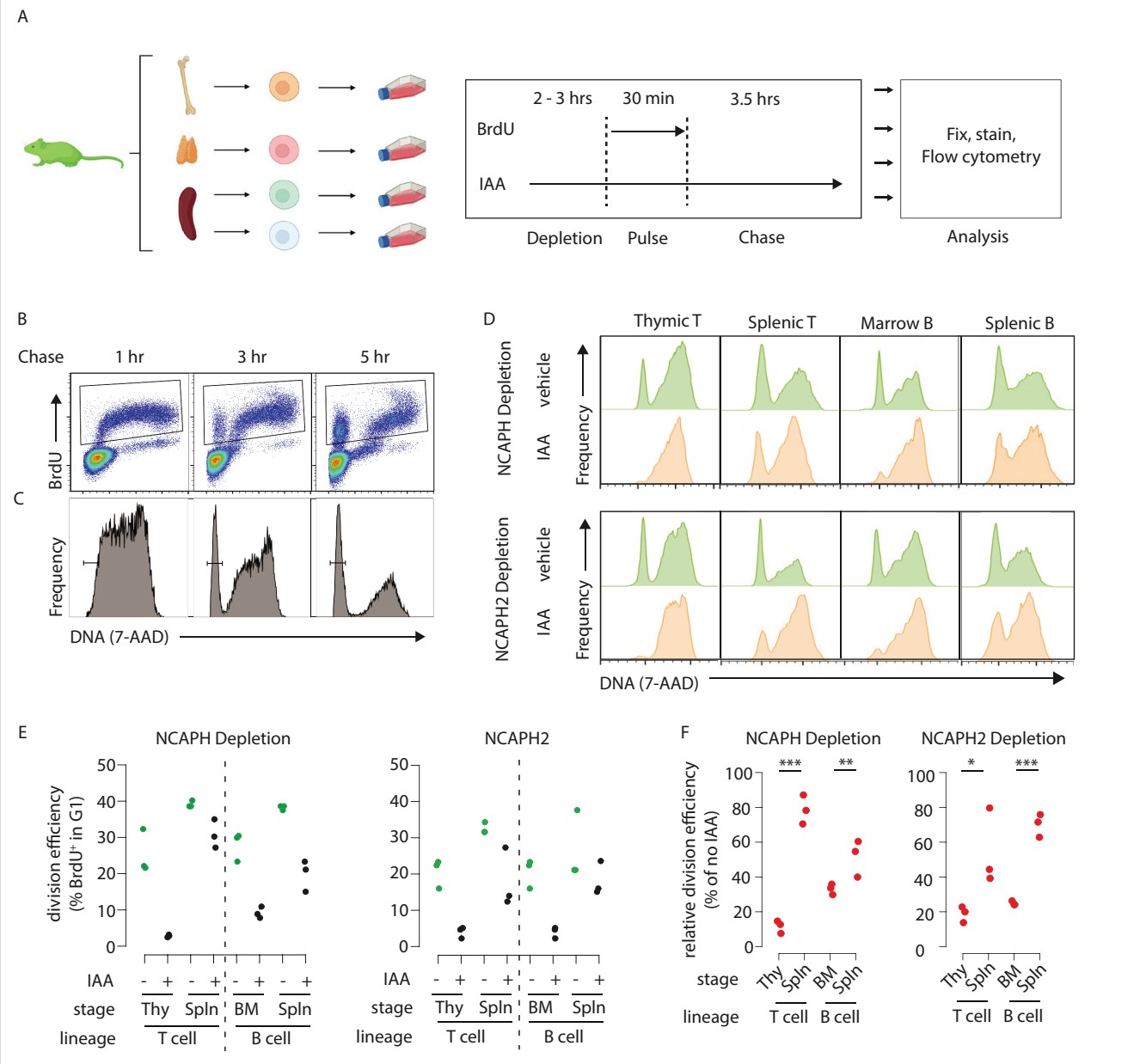

**Figure 5.** Dynamic changes in condensin dependency during lymphocyte differentiation. (**A**) Chronological representation of the BrdU pulse chase assay to measure the efficiency of cell division in primary cell types cultured ex vivo. Lymphocyte isolation and culture protocols are detailed in Materials and methods. Quantifying the % of BrdU⁺ cells (**B**) that complete mitosis and halve their DNA content (**C**) allows the efficiency of a single-cell division to be quantified under normal or acute condensin deficient conditions. The appearance of BrdU⁺G1 cells can be seen at 3 and 5 hr. (**D**) Representative DNA content profiles, gated on BrdU⁺ as shown in panel B, from cycling early (thymic/marrow) or activated mature (Splenic) T and B lymphocytes, measured following a 3.5-hr chase in the presence or absence of condensin I or II. (**E**) Quantification of division efficiency, based on the % of BrdU⁺ cells in G1 after 3.5 hr (n = 3 biological replicates from at least 2 independent experiments). Corresponding condensin depletion levels for each experiment are shown in *Figure 5—figure supplement 1C* (**F**). Quantification of the effect of NCAPH or NCAPH2 degradation on cell division across cell types in panel E. For each cell type, division efficiency (panel E) in the vehicle only control condition was set to 100%, and the same parameter in IAA-treated cells was expressed relative to this. Asterisks represent p values from paired *t*-tests ***p < 0.01, **p < 0.05, *p < 0.1.

The online version of this article includes the following figure supplement(s) for figure 5:

**Figure supplement 1.** Dynamic changes in condensin dependency during lymphocyte differentiation.

following I.P. injection of IAA or vehicle. No significant differences were observed (*Figure 6—figure supplement 1B*).

Adult animals homozygous for AID-tagged kleisin alleles and *Rosa26^Tir1* were then injected I.P. with IAA, then haematopoietic organs were collected either 1 or 2 hr post-injection (*Figure 6A*). Flow cytometric quantification of Clover fluorescence in proliferating thymocytes (*Figure 6A*) and bone marrow B-cell precursors (*Figure 6—figure supplement 1C*) revealed that a majority of target protein was typically degraded within 1 hr of injection, and near complete degradation (>90%) was achieved within 2 hr, although some variability was observed between biological replicates. To validate knock-down efficiency using an orthogonal method, and to assess the proteome-wide specificity of the AID system, thymus tissue was collected from *Ncaph2^{AID:Clover/AID:Clover} Rosa26^{Tir1/Tir1}* animals 2 hr following I.P. injection of IAA or vehicle, and proteomic quantification was performed in MACS-purified CD8⁺ cells using mass spectrometry (*Figure 6B*). This confirmed profound (~10-fold) downregulation of the target protein. Remarkably, no other protein was significantly downregulated using thresholds of p < 0.01 and >2-fold change. Only a single protein (the heat shock protein Hspb11) was significantly upregulated. Relaxing the significance threshold to p < 0.05 led to only three downregulated proteins and an additional two upregulated proteins (*Supplementary file 1*). We conclude that IAA injection can achieve not only rapid and profound, but also highly specific degradation of AID-tagged proteins in vivo. However further proteomic studies are needed to assess the potential for TIR1-dependent off-target protein degradation in other cell types, and in combination with other target proteins.

To assess the kinetics of protein degradation and recovery over longer periods following single dose I.P. administration of IAA, we generated *Ncaph^{AID:Clover/+} Rosa26^{Tir1/Tir1}* animals, in which AID-tagged protein could be degraded while leaving a pool of untagged protein to support ongoing cell division. Protein levels began to recover within 6 hr post-injection before returning to baseline levels within 72 hr (*Figure 6C*). This shows that degradation of endogenous proteins via the AID system is reversible in vivo.

Target protein was also efficiently degraded in 2 hr within mitotic crypt cells of the small intestine (*Figure 6—figure supplement 2A*). However, we observed individual interphase cells that appeared resistant to degradation (*Figure 6—figure supplement 2A*, white arrow). Similarly, bone marrow erythroblasts (TER119⁺) retained Clover fluorescence after IAA exposure in vivo (*Figure 6—figure supplement 2B*). TIR1 protein was robustly expressed in these cells (*Figure 6—figure supplement 2C*), ruling out cell-lineage-specific transgene activity as the source of cell-lineage-specific IAA pharmacodynamics. Nonetheless, the barrier to IAA activity in erythroblasts was cell intrinsic rather than a property of the tissue environment, because TER119⁺ cells in ex vivo bone marrow cultures also failed to degrade NCAPH in response to IAA treatment, whereas CD19⁺ cells in the same culture degraded efficiently (*Figure 6—figure supplement 2D*).

Protein degradation was also inefficient in dividing spermatocytes (*Figure 6—figure supplement 2E*). In this case, we speculate that the blood–testes barrier could prevent IAA from entering seminiferous tubules to effect protein degradation, although cell intrinsic barriers to degradation cannot be excluded.

Finally, we tested the ability of IAA to degrade NCAPH^{AID:Clover} protein in embryos. Adult females homozygous for *Ncaph^{AID:Clover}* and *Rosa26^{Tir1}* transgenes were mated with males of the same genotype, then injected I.P. with IAA (100 mg kg⁻¹) or vehicle at 10.5 days post coitum (*Figure 6D*). Embryos were collected 4 hr later, and NCAPH levels were quantified by immunofluorescence performed on whole-mount embryonic cryosections co-stained with antibodies recognising E-Cadherin (CDH1). This revealed near complete protein degradation within CDH1-positive regions of the developing surface ectoderm (*Figure 6E–G*). Similarly, profound degradation was observed within PDGFR⁺ cells of the embryonic mesenchyme (*Figure 6—figure supplement 1D, E, F and G*). These data show that IAA is able to cross the placenta to achieve robust protein degradation in embryonic cells.

## Discussion

In this paper, we describe an approach that opens up new possibilities for studying the consequences of acute protein loss in mammalian primary cells, tissues and whole organisms (*Figure 6—figure supplement 3*). The ability to trigger protein degradation using small molecules has numerous advantages over alternative reverse genetic approaches. Most importantly, protein function is removed in less than 2 hr; substantially quicker than would be possible using gene editing nucleases, recombinases,

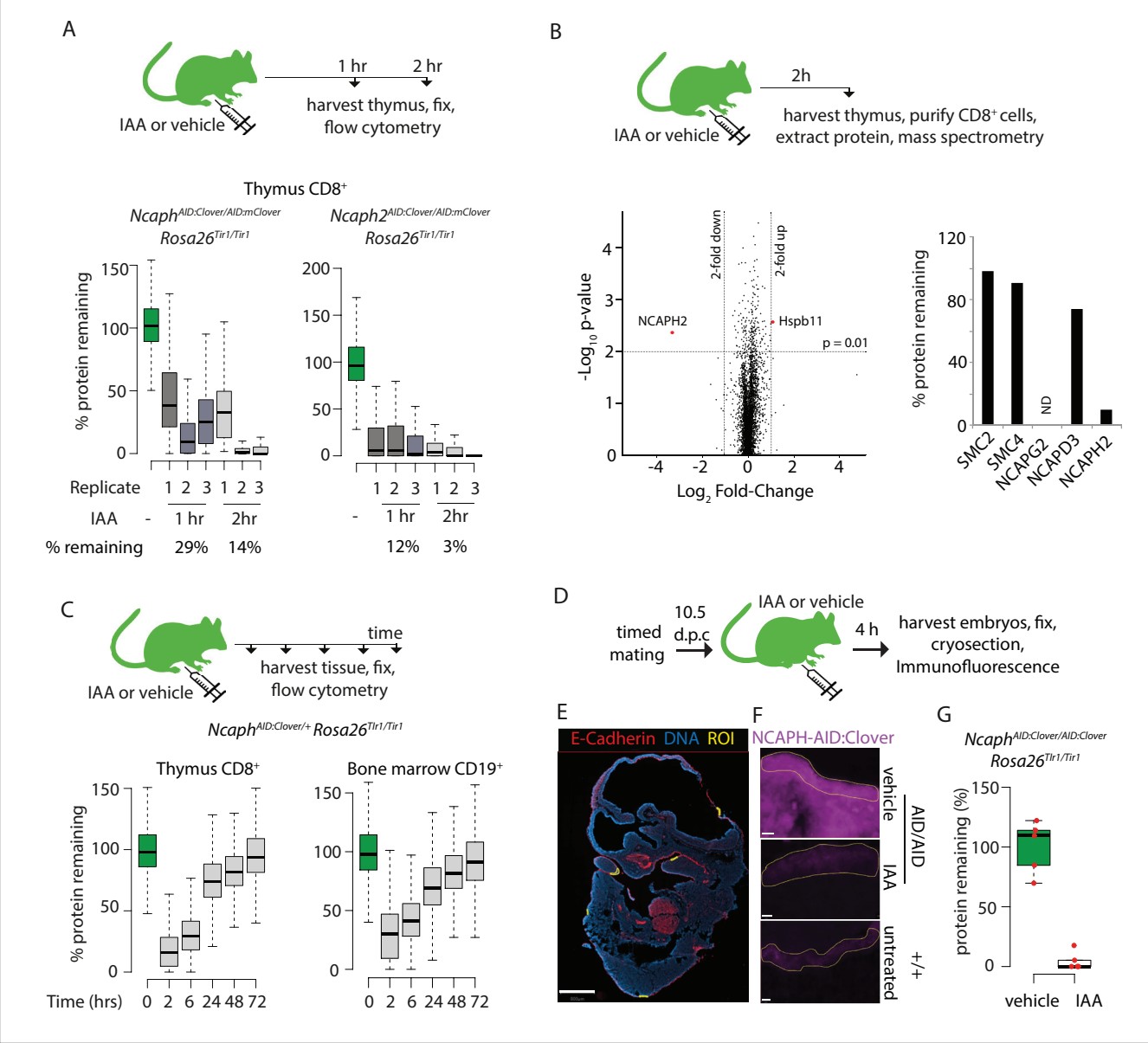

**Figure 6.** Rapid degradation of endogenous tagged proteins in living mice. (**A**) (Top) I.P. injection time course to test protein degradation in vivo. Each mouse received a single injection of IAA solution (100 mg/kg), or vehicle. (Bottom) Boxplots show the extent of targeted protein degradation in >1000 S/G2/M CD8⁺ thymocytes harvested 1 or 2 hr following auxin injection, measured by flow cytometry. % protein remaining was calculated as described in *Figure 2* legend. Boxes indicate the boundaries of upper and lower quartiles and whiskers show the range. Data are from three biological replicate injections performed over at least two independent experiments. (**B**) Proteome quantification by mass spectrometry analysis of MACS-purified CD8⁺ thymocytes. *n* = 3 animals per condition. (**C**) Protein degradation and recovery following a single I.P. injection. Data are presented as described for panel A, except mice were heterozygous for the *Ncaph2*^AID:Clover^ allele. (**D**) Schematic illustration of experimental workflow for protein degradation in E10.5 embryos. (**E**) Example image from whole-mount immunofluorescence performed on E10.5 embryo cryosections, stained with DAPI, anti-GFP-647 nanobooster (detecting NCAPH-AID:Clover), and anti-CDH1. Anti-GFP signal was quantified within five CDH1⁺ regions of interest (ROI) per embryo, which were selected based solely on the CDH1 staining pattern. To enable CDH1 localisation and ROIs to be visualised, the anti-GFP-647 channel is not shown in this panel. Images were captured at ×40 magnification, scale bar = 800 µm. (**F**) Example ROI's from CDH1⁺ stained tissue on which target protein quantification was performed. To visualise degradation, only the NCAPH-AID:Clover channel is shown. Scale bar = 10 µm. (**G**) Quantification of degradation efficiency in CDH1⁺ embryonic cells. Mean pixel intensity was first calculated from five Cdh1⁺ regions in *Ncaph*^AID:Clover/AID:Clover^ *Rosa26*^Tir1/Tir1^ embryos from mothers injected with either IAA or vehicle, and non-fluorescent negative control embryos (*n* = 1 embryo each). The mean pixel intensity value from negative control ROIs was set to 0%, and the mean value from vehicle-only ROIs to 100%. Mean pixel intensity values for each ROI from vehicle and IAA-exposed embryos were then plotted on this scale. Negative values were set to 0%.

The online version of this article includes the following figure supplement(s) for figure 6:

*Figure 6 continued on next page*

*Figure 6 continued*

**Figure supplement 1.** Rapid degradation of endogenous tagged proteins in living mice.

**Figure supplement 2.** Rapid degradation of endogenous tagged proteins in living mice.

**Figure supplement 3.** General schematic and timeline for generating germline transgenic mice for studying protein function using the auxin-inducible degron (AID) system.

or RNAi. Rapid removal is particularly important for studying essential cell cycle proteins such as condensins, where secondary and tertiary phenotypes arising downstream from abnormal cell division can quickly obscure primary phenotypes (*Hocquet et al., 2018*). Given the central importance of condensins in establishing chromosome architecture during cell division (*Ganji et al., 2018*; *Gibcus et al., 2018*; *Hirano et al., 1997*; *Ono et al., 2003*), combined with persistent and unresolved questions about their involvement in other physiological processes in mammals (*Dowen et al., 2013*; *Rawlings et al., 2011*; ; *Swygert et al., 2019*; *Wu et al., 2019*), we anticipate that the two mouse models described here will provide an important resource for the chromosome biology community.

To demonstrate the utility of this system, we compared the ability of non-immortalised primary lymphocyte populations, from different lineages and stages of differentiation, to undergo a single round of cell division in the near absence of Ncaph and Ncaph2 (*Figure 5*). Our results show that different cell types differ in their ability to complete a single round of cell division in the near absence of these proteins, adding to an increasing body of data suggesting cell-type-specific differences in the consequences of condensin perturbation (*Elbatsh et al., 2019*; *Gosling et al., 2007*; *Martin et al., 2016*; *Woodward et al., 2016*). Elucidating the cause of these differences was beyond the scope of the current study, but it could potentially involve cell-type-specific chromosome topology (e.g. fewer catenations to remove in mature cells), or cell cycle checkpoints (e.g. greater sensitivity to mitotic arrest in precursor cells). In the future, we expect these two degrader mouse lines will provide valuable insights into chromosome biology and mitotic chromosome structure in the context of in vivo cellular heterogeneity, development, and disease.

Protein degradation was achievable in a range of cell types, including B and T lymphocytes at different developmental stages (precursor and mature, *Figure 5—figure supplement 1C*), fibroblasts (*Figure 2D*), neural stem cells (*Figure 2E*), gut epithelial cells (*Figure 6—figure supplement 2A*), embryonic CDH1$^+$ cells of the developing surface ectoderm (*Figure 6G*), and PDGFR$^+$ cells of the embryonic mesenchyme (*Figure 6—figure supplement 1G*). However, a minority of cell types proved refractory. For example, erythroid progenitors failed to degrade AID-tagged proteins due to a cell intrinsic block (*Figure 6—figure supplement 2B* and *Figure 5—figure supplement 1D*) that was not attributable to silencing of the *Tir1* transgene (*Figure 6—figure supplement 2C*). We speculate that erythroblasts fail to express one or more endogenous subunits of the SCF$^{Tir1}$ E3 ubiquitin ligase complex, or another component of the ubiquitin proteosome system that is necessary for degradation via the AID system. Spermatocytes also failed to degrade (*Figure 6—figure supplement 2D*), possibly due to poor ligand transit across the blood testes barrier, poor expression of TIR1 or required endogenous factors. An analogous barrier prevents many small molecules from penetrating into the adult brain. Poor expression of condensins in post-mitotic tissues prevented us from testing AID function in the brain; however, a structurally related ligand for the AID2 system (5'Ph-IAA) was found to have relatively weak protein degradation activity in this tissue (*Yesbolatova et al., 2020*).

By combining ligand titrations with genetic crosses to generate animals bearing different allelic combinations, we showed that the dosage of all three components of the ternary complex formed between the target protein, ligand and TIR1 substrate receptor protein, can determine the kinetics of protein degradation in mammalian primary cells (*Figures 2E–4*). Specifically, reduced expression of TIR1 (*Figure 3*, *Figure 3—figure supplement 1*), or increased expression of degron-tagged protein (*Figure 4*), can both decrease the efficiency of protein degradation. Notably, reduced expression of the CRBN substrate receptor, and increased expression of non-essential neosubstrates, have emerged as mechanisms through which haematological malignancies can develop resistance to thalidomide analogues (*Heintel et al., 2013*; *Sperling et al., 2019*). Saturation of protein degradation activity is therefore a general property of molecular glue compounds, which might limit their activity on highly expressed targets and influence protein degradation efficiency in both clinical and research contexts.

Concerns have been raised about the AID system because a subset of AID-tagged proteins can undergo TIR1-dependent turnover even in the absence of exogenous IAA. We showed that the

presence of TIR1 had little or no effect on the expression of two different AID-tagged condensin subunit proteins in the absence of ligand (*Figure 1G*), but enabled their rapid degradation, often to levels below 10% of baseline, within 1–2 hr of ligand exposure (*Figures 2 and 6*). Regulated induction of TIR1 expression in yeast has demonstrated that leaky degradation of target proteins occurs as a consequence of TIR1 overexpression (*Mendoza-Ochoa et al., 2019*). Achieving TIR1 expression levels sufficient for IAA-inducible degradation yet insufficient for IAA-independent degradation is therefore of paramount importance. That leaky degradation did not cause problems in our study suggests that the single-copy *Rosa26^{Tir1}* allele generated via genome editing is expressed within this desirable range, at least for these two proteins. Further work is required to determine whether other AID-tagged targets show leaky degradation in combination with the *Rosa26^{Tir1}* allele generated here.

While our work was ongoing, point mutations in TIR1 (F74A and F74G) have been identified that, in combination with a modified ligand, eliminate TIR1-dependent leaky degradation, and allow degradation to be induced at much lower ligand concentrations (*Nishimura et al., 2020*; *Yesbolatova et al., 2020*). We therefore predict that AID tagging should be applicable to study a broad range of intracellular proteins in mice and their primary cell derivatives.

A very recent publication has reported targeted degradation of endogenously tagged NELFB protein using dTAG (*Abuhashem et al., 2022*): another promising degron-tagging approach (*Nabet et al., 2018*; *Nabet et al., 2020*). dTAG uses distinct tags, ligands, and E3 ligase interactions, and should therefore be compatible with the AID system developed here to enable orthogonal chemical control of two different proteins in mice. In contrast to AID, which requires exogenous expression of the E3 ligase protein TIR1, dTAG uses endogenous E3 ligases to achieve protein degradation and could therefore be simpler to establish in mice. However, this same property could also prevent future applications that require tissue- or cell-type selective protein degradation. This is achievable using AID via tissue-specific expression of TIR1, as shown in *C. elegans* (*Ashley et al., 2021*; *Zhang et al., 2015*). Although not tested here, the Rosa26 loxP-STOP-loxP TIR1 allele generated during this study could allow spatial restriction of IAA pharmacodynamics to cell types co-expressing Cre recombinase.

Although the AID system presents a few important advantages over more traditional conditional genetic systems such as Cre-loxP, including speed and reversibility, several issues remain to be addressed after our proof-of-concept study. First, AID and other degron-tagging approaches require genetic fusion of the tag to an endogenous protein. Compared to the intronic loxP sites for the Cre-loxP system, it is more likely that protein tags will interfere with normal protein function (e.g. *Figure 1—figure supplement 1B, C*). The effect of a fusion tag is highly dependent on the unique properties of the target protein and needs to be analysed and validated on a case-by-case basis. Second, the toxicity profile of IAA in rodents has only been characterised in a limited manner (*Furukawa et al., 2005*; *Ji et al., 2019*), in contrast to Tamoxifen: the most broadly used chemical induction reagent for the Cre-loxP system. Although we did not observe overt toxicity following single dose administration of IAA at 100 mg/kg in our study, further work is needed to characterise the potential toxicities. Third, the potential 'leaky' degradation of proteins by TIR1 without IAA administration needs to be properly controlled.

At this point, three non-mutually exclusive hypotheses can be speculated for the mechanism of leaky degradation. First, the leaky degradation may be an intrinsic property of the TIR1 protein. As described above, the AID2 system uses mutations in the TIR1 protein and a chemically modified ligand, which were able to minimise leaky degradation for several proteins in cultured cells (*Yesbolatova et al., 2020*). Given that our study used the original AID system and did not observe substantial leaky degradation (*Figure 1G*), further work is warranted to establish whether AID or AID2 is intrinsically more leaky in mice. Second, the leakiness, and indeed the efficiency, of auxin-inducible degradation may be dependent on the structural and biochemical properties of each target protein, as recently observed with PROTAC degradation systems (*Bondeson et al., 2018*). Third, dietary factors might promote leaky degradation. Although all mice in this study were fed on standard plant-based chow, plant-based foods could theoretically introduce more IAA-like molecules than, for example, casein-based diets. The gut microbiome might also play a role in producing IAA-like molecules (*Lai et al., 2021*). Thus, detailed knowledge of the housing and feeding condition of mouse colonies might be an important factor for the precise reproduction of experiments using AID systems.

In conclusion, these issues need to be resolved in order to establish AID as a general system for conditional genetic manipulation for the broader mouse genetics community. Nonetheless, our

demonstration that degron tagging systems are effective in living mice should enable more versatile conditional alleles to study protein function, and to model drug activity, in mouse models of development and disease.

# Materials and methods

**Key resources table**

| Reagent type (species) or resource | Designation | Source or reference | Identifiers | Additional information |
|---|---|---|---|---|
| Gene (*O. sativa*) | *osTIR1* | Addgene | 64,945 | |
| Gene fragment (*A. thaliana*) | *mAID* | **Nora et al., 2017** | IAA17 | |
| Strain, strain background (*Mus musculus*) | CD1 | Charles River | Crl:CD1(ICR) | |
| Strain, strain background (*Mus musculus*) | C57BL/6J | Charles River | | |
| Genetic reagent (*Mus musculus*) | *Rosa26-LSL-osTIR1-9myc* | This paper | CD1-Gt(ROSA)26Sorem1(CAG-LSL-osTIR1-myc)Jrt | Sequence details in **Supplementary file 3** |
| Genetic reagent (*Mus musculus*) | *Rosa26-osTIR1-9myc (Mus musculus)* | This paper | CD1-Gt(ROSA)26Sorem1.1(CAG-LSL-osTIR1-myc)Jrt | Sequence details in **Supplementary file 3** |
| Genetic reagent (*Mus musculus*) | *pCX-NLS-cre (Mus musculus)* | **Belteki et al., 2005** | ICR-Tg(CAG-cre) | http://www.informatics.jax.org/reference/J:99607 |
| Genetic reagent (*Mus musculus*) | *Ncaph$^{AID:Clover}$* | This paper | *Ncaph$^{AID:Clover}$* | Sequence details in **Supplementary file 3** |
| Genetic reagent (*Mus musculus*) | *Ncaph2$^{AID:Clover}$* | This paper | *Ncaph2$^{AID:Clover}$* | Sequence details in **Supplementary file 3** |
| Recombinant DNA reagent | pRosa26-CAG-LSL-osTIR1 | This paper | | **Belteki et al., 2005** |

## Mouse maintenance and husbandry

All animal work was approved by a University of Edinburgh internal ethics committee and was performed in accordance with institutional guidelines under license by the UK Home Office. AID knock-in alleles were generated under project license PPL 60/4424. *Rosa26$^{Tir1}$* knock-in mouse lines were generated under the Canadian Council on Animal Care Guidelines for Use of Animals in Research and Laboratory Animal Care under protocols approved by the Centre for Phenogenomics Animal Care Committee (20-0026H). Experiments involving double transgenic animals were conducted under the authority of UK project license PPL P16EFF7EE.

Mice were maintained at the Biological Research Facility, Western General Hospital, Edinburgh and fed on RM3 chow (Special Diets Services; Product Code 801700). All experimental animals were between 6 and 16 weeks in age unless otherwise specified. Male and female animals were used interchangeably. Mice were housed in individually ventilated cages with 12 hr light/dark cycles. All tissues were harvested and processed immediately following euthanasia via cervical dislocation or $CO_2$. All animals and primary cells used in protein degradation experiments were of a mixed (C57BL/6J and CD1) genetic background.

## Generation of Ncaph and Ncaph2 degron-reporter mice

*Ncaph-AID-mClover* and *Ncaph2-AID-mClover* mice were generated following the Easi-CRISPR protocol (**Miura et al., 2018**). sgRNAs were designed using the Zhang Lab design tool (https://zlab.bio/guide-design-resources) and ordered from IDT. Priority was given to protospacer sequences that would result in cleavage proximal to the stop codon with low predicted likelihood for off-target cleavage. Repair templates were long single stranded oligonucleotides ('megamers') ordered from IDT. Each megamer comprised 105 nucleotides of homology either side of the integrated sequence. Integrations included linker sequence and the 44 amino acid mini-auxin-inducible degron used by Nora et al. (**Morawska and Ulrich, 2013**; **Nora et al., 2017**) fused in-frame with the fluorescent protein Clover (**Lam et al., 2012**). Full nucleotide sequences for the guide RNA target sequences and repair templates are listed in **Supplementary file 3**.

The microinjection mix comprised pre-annealed crRNA/TracrRNA complex (20 ng/µl), repair template (5 ng/µl), and Cas9 protein (NEB – 0.3 µM). This was incubated at 37°C for 10 min before microinjection. Zygotes were collected from C57BL/6J females mated overnight with C57BL/6J stud males (Charles River Laboratories). Editing reagents were introduced via pronuclear microinjection at the Evans Transgenic Facility (University of Edinburgh), cultured overnight before transfer to pseudo-pregnant CD1 females. Successful integrations were identified by PCR using primers spanning the integration sites, then confirmed by observing band shifts on western blots probed with antibodies against wildtype NCAPH and NCAPH2 (*Figure 1C*). Founder animals were outcrossed for two generations with C57BL/6J animals and then N2 siblings were intercrossed to obtain homozygotes.

## Generation of *Rosa26^Tir1^* knock-in mice

The plasmid donor for generating *Rosa26^Tir1^* knock-in mice was constructed as follows. A TIR1-9myc cassette was PCR amplified from the pBABE-TIR1-9myc plasmid (addgene 64945, a kind gift from Don Cleveland *Holland et al., 2012*). The cassette was inserted into MluI restriction site of the pR26 CAG AsiSI/MluI plasmid (addgene 74286, a kind gift from Ralf Kuehn, *Chu et al., 2016*) by infusion cloning (Takara). The coding sequence of TIR1-9myc was separated from the CAG promoter by a loxP-STOP-loxP (LSL) cassette.

The *Rosa26-LSL-osTIR1-9myc(CD1-Gt(ROSA)26Sor^em1(CAG-LSL-osTIR1-myc)Jrt^)* mouse line was generated using the 2C-HR-CRISPR method (*Gu et al., 2018*). Cas9 mRNA (100 ng/µl), R26 sgRNA (50 ng/µl), and the Rosa26-LSL-osTIR1-9myc circular donor plasmid (30 ng/µl) were microinjected into 2-cell stage embryos of CD1 mice. Full nucleotide sequences for the guide RNA target sequences and repair templates are listed in *Supplementary file 3*. Embryos were cultured overnight in KSOMaa (Cytospring) to reach the morula stage and then transferred to pseudopregnant CD1 females. Fifteen live pups were produced, three of them were validated to contain the correct insert by long range PCR. One male founder was crossed with a wildtype CD1 female to generate N1 offsprings. The N1s were first screened by long range PCR. For the positives, knock-in junction sequences were validated by sanger sequencing. Droplet Digital Quantitative PCR (ddqPCR) was performed by the The Center for Applied Genomics (TCAG) in Toronto to measure the copy number of the insert. All N1 animals tested had single-copy insertions. The mouse line was outcrossed for another three generations to remove any possible off-target mutations that might have been induced by genome editing. The Rosa26-LSL-osTIR1-9myc mouse line is homozygous viable and were kept as homozygous breeding.

To generate the *Rosa26^Tir1^* allele (full name *Rosa26-osTIR1-9myc (CD1-Gt(ROSA)^26Sorem1.1(CAG-LSL-osTIR1-myc)Jrt^*) used in subsequent experiments, a *Rosa26-LSL-osTIR1-9myc ^+/−^* male was mated with a *pCX-NLS-cre (ICR-Tg(CAG-cre)^1Nagy^* female. Progeny were screened for removal of the LSL cassette by PCR and the mice carrying correctly recombined sequences were bred to establish the line. The *Rosa26^Tir1^* allele is homozygous viable and mice were bred in the homozygous state.

## Whole cell protein extract preparation and quantification

Protein preparations were generated from either single-cell suspensions of primary haematopoietic cells, or whole tissue. Single-cell suspensions of thymus were generated by gentle dissociation of whole thymus tissue through 40 µm filters (Fisherbrand, 22-363-547). For bone marrow, tissue was flushed out of tibia and femur bones with phosphate-buffered saline (PBS) before dissociation through a 40 µm filter. Cell numbers were counted manually using a haemocytometer. Bone marrow cells were further purified by Magnetic Activated Cell Sorting (Miltenyi), using beads pre-coated with antibodies against B220, Ter-119, CD4, or CD8. MACS purification proceeded according to the manufacturer's instructions. Cell pellets were resuspended in NP-40 Lysis Buffer (150 mM NaCl, 50 mM Tris–HCl, 1% NP-40) using 3 volumes of NP-40 Buffer to 1 volume of cell pellet. 0.5 µl benzonase nuclease (Millipore) was added per 100 µl of resuspended pellet. Samples were incubated at 4°C for 30 min with intermittent vortexing, before pelleting cellular debris via centrifugation at maximum speed (13,200 × *g*, 15 min, 4°C). For whole tissue samples, adult tissue (brain, thymus, lung, spleen, kidney, small intestine, and liver) were removed, snap frozen in LN2, and stored at −80°C until use. Between 10 and 30 mg of frozen tissue was weighed and homogenised in 1 ml RIPA buffer (150 mM NaCl, 1% NP-40, 0.5% NaDeoxycholate, 0.1% SDS, 50 mM Tris–HCl pH 8 with 5 µl benzonase [Millipore]) for 10 min using a TissueLyserLT (Qiagen), then incubated on ice for 30 min. Cellular debris was pelleted via centrifugation at maximum speed for 15 min at 4°C. Supernatants were transferred into fresh

tubes, and protein concentration was quantified using a Pierce BCA Protein Assay Kit (Thermo, 23228) following the manufacturer's instructions.

Pierce Lane Marker Reducing Sample Buffer (1×, Thermo, 39000) was added to each sample prior to denaturation via boiling at 95°C for 5 min. Samples were used immediately or stored at −20°C.

## Western blotting

Denatured protein lysates (12.5 µg/sample) were loaded on to NuPAGE 4–12% Bis–Tris 1.0 mm Mini Protein Gels (Invitrogen, NP0321) alongside Chameleon Duo Protein Ladder (3 µl/lane; LiCOR, 928-60000) or PageRuler Protein Ladder (5 µl/lane; Thermo Scientific, 26616) and run in pre-chilled 1× MOPS Buffer (Thermo, NP0001). Samples were typically run at 100 V for 90 min. Transfers were performed using either the iBlot2 Gel Transfer device according to the manufacturer's instructions or wet transfer. PVDF membranes were pre-soaked in 100% methanol (Fisher, 10284580) and rinsed briefly in Transfer Buffer (25 mM Tris [AnalaR, 103156X], 200 mM glycine [Fisher, G-0800-60], 20% methanol, 0.02% SDS [IGMM Technical Services]). Genie Blotter transfer device (Idea Scientific) was assembled with the gel and PVDF membrane placed between two layers of cellulose filter paper (Whatman, 3030-917) inside the loading tray. Once the apparatus was prepared, Transfer Buffer was filled to the top of the Genie Blotter and transfer proceeded for 90 min at 12 V.

Conditions for blocking and antibody staining were optimised individually for each probe. Samples were blocked with either 5% milk powder (Marvel) in Tris Buffered Saline (IGMM Technical Services) with 0.1% Tween-20, or 3% bovine serum albumin (BSA) (Sigma) in TBS with 0.1% Tween-20, with constant agitation, either at room temperature for 1 hr, or at 4°C overnight.

Primary antibodies were added to the corresponding block solution at the dilution shown in *Supplementary file 2*. Membranes were incubated in the antibody dilutions with constant agitation, either at room temperature for 1 hr, or at 4°C overnight. Membranes were washed in TBS-Tween-20 solutions (0.1% Tween-20; four washes × 10 min). Fluorescent or HRP-conjugated secondary antibodies were also diluted in the corresponding block solution (with 0.1% Tween-20), and membranes were incubated with secondary antibody dilutions under constant agitation at room temperature for 1 hr. Membranes were then washed in TBS-Tween-20 solutions (0.1% Tween-20, four washes × 10 min). Membranes were visualised on an Odyssey CLx Imaging System (LiCOR) or ImageQuant (Cytiva). Fluorescent antibodies were detected using either a 700 Channel Laser Source (685 nm) or 800 Channel Laser Source (785 nm).

## Proteome analysis by mass spectrometry

Mice were treated with 100 mg/kg auxin via I.P. injection for 2 hr, then culled by cervical dislocation. Thymus was removed and a single-cell suspension of primary thymocytes was made using ice-cold PBS. CD8a$^+$ cells were isolated by MACS using CD8a2 (Ly-2) microbeads (Miltenyi) following the manufacturer's instructions. Purified cells were lysed in whole proteome lysis buffer (6 M GuHCl, 100 mM Tris–HCl 8.5, 1 mg/ml chloracetamide, 1.5 mg/ml TCEP) at a concentration of $0.3 \times 10^6$ cells/µl buffer. Lysate was sonicated with a probe sonicator (Soniprep 150) until no longer viscous, and boiled at 95°C for 5 min, then centrifuged at 14,000 rpm for 5 min. Supernatent was then transferred to a fresh tube and processed for mass spectrometry.

A 50 µl volume of sample was heated to 97°C for 5 min, then pre-digest (Lys-C, Fujifilm WakoPure Chemical Corporation) was added (1 µl/sample) and samples incubated at 37°C for 3 hr. Samples were diluted 1/5 by addition of 200 µl Mass Spec grade water, and 1 µg trypsin (Fujifilm WakoPure Chemical Corporation) was added to each sample. Samples were incubated at 37°C with hard shaking overnight.

Samples were acidified with 1% Trifluoroacetic acid (TFA) and centrifuged at 13,000 rpm, for 10 min at room temperature. Sample was applied to a double-layer Empore C18 Extraction Disk (3 M) prepared with methanol. Membrane was washed twice with 0.1% TFA and protein was eluted with elution buffer (50% acetonitrile (ACN), 0.05% TFA), dried using a CentriVap Concentrator (Labconco) and resuspended in 15 µl 0.1% TFA. Protein concentration was determined by absorption at 280 nm on a Nanodrop 1000, then 2 µg of de-salted peptides were loaded onto a 50 cm emitter packed with 1.9 µm ReproSil-Pur 200 C18-AQ (Dr Maisch, Germany) using a RSLC-nano uHPLC systems connected to a Fusion Lumos mass spectrometer (both Thermo, UK). Peptides were separated by a 140-min linear gradient from 5% to 30% acetonitrile, 0.5% acetic acid. The mass spectrometer was operated in

DIA mode, acquiring a MS 350–1650 Da at 120 k resolution followed by MS/MS on 45 windows with 0.5 Da overlap (200–2000 Da) at 30 k with a NCE setting of 27. Raw files were analysed and quantified using Spectronaut 15 (Biognosis, Switzerland) using directDIA against the Uniprot *Mus musculus* database with the default settings. Ratios and *t*-tests were calculated by the Spectronaut pipeline using default settings.

## Flow cytometry

For cultured adherent cells, single-cell suspensions were first generated using trypsin (MEFs) or accutase (neural stem cells). For haematopoietic cells, samples were prepared from single-cell suspensions of bone marrow and thymus. Samples were incubated with fluorescently conjugated antibodies against cell surface markers (*Supplementary file 2*) and Fixable Viability Dye (eBioscience, 65-0865-14, 1 in 200 dilution) diluted in Flow Cytometry Staining Buffer (eBioscience, 00-4222-26) (20 min at 4°C). Samples were then washed in a 10-fold volume of Flow Cytometry Staining Buffer before centrifugation at 300 × *g* for 5 min at 4°C. Pellets were resuspended in Cytofix/Cytoperm solution (BD Bioscience, 554722) following the manufacturer's instructions and washed in Perm/Wash buffer (BD Bioscience, 554723). If required, samples were incubated with fluorescently conjugated antibodies against intracellular markers for 20 min at room temperature. For intracellular γH2AX staining, samples were further permeabilised by resuspending in Perm/Wash buffer (1 ml) for 15 min at 4°C before antibody incubation. After intracellular antibody incubation, all stained samples were then washed in Perm/Wash buffer (300 × *g*/5 min/4°C). Cell Trace Yellow (Thermo Fisher C34567) experiments were conducted according to the manufacturer's protocol. Samples were resuspended in DAPI staining solution (1 µg/ml DAPI in PBS). DAPI-stained samples were incubated on ice for at least 15 min before data acquisition.

Data acquisition (BD LSRFortessa) was performed no more than 24 hr following sample fixation. Identical laser power was used to quantify Clover signal across all experiments. Data analysis was conducted using FlowJo software (Treestar). Cellular debris/aggregates were excluded using strict forward- and side-scatter gating strategies. Cell cycle stages were gated based on DNA content (DAPI) fluorescence. Our protein degradation experiments focused on S/G2/M phase cells in order to control for cell cycle differences between cell types, and because condensins function primarily during cell division. Wildtype samples lacking Clover expression were processed and stained in parallel to transgenic samples. To correct for autofluorescence, background fluorescence was measured for each cell population from wildtype samples, and then subtracted from transgenic fluorescence values. To generate boxplots, the background-corrected fluorescence value from each of >1000 cells was expressed relative to the mean of the vehicle only condition. We focused exclusively on S/G2/M cells, gated on DNA content, for quantifications to avoid the confounding effects of quiescent cells, where condensins are expressed at very low levels.

## Primary cell culture

### Thymic and bone marrow ex vivo cultures

Single-cell suspensions of thymus tissue were generated by gentle dissociation of whole thymus tissue through 40 µm filters (Fisherbrand, 22-363-547) into PBS. For bone marrow, tissue was flushed out of tibia and femur bones with PBS before dissociation through a 40-µm filter. Cell numbers were counted manually using a haemocytometer. Cells (1–2.5 × 10⁶/ml) were then cultured at 37°C ex vivo for 2–6 hr in RPMI (Gibco, 21875-034) containing 10% FCS (Fetal Calf Serum: IGC Technical Services) and penicillin (70 mg/l, IGC Technical Services) and streptomycin (130 mg/l, IGC Technical Services). For bone marrow cultures, different cell lineages were cultured together and then B cell and erythroid lineages were identified based on flow cytometric detection of cell surface marker expression (CD19 and Ter119, respectively) and analysed separately.

## Peripheral T and B lymphocyte ex vivo cultures

Peripheral T and B lymphocytes were derived from spleens dissected from adult animals. Single-cell suspensions of splenic tissue were generated by gentle dissociation of whole spleen through 40 µm filters (Fisherbrand, 22-363-547). Splenic cells were resuspended in MACS buffer (0.5% BSA (Sigma), 1 mM EDTA in PBS – 40 µl MACS Buffer per 10 × 10⁷ cells) in preparation for Magnetic Activated Cell Sorting. Peripheral T and B cells were isolated from whole spleen using Pan T Cell (Miltenyi Biotec,

130-095-130) or Pan B Cell (Miltenyi Biotec, 130-104-433) isolation kits, respectively, according to the manufacturer's instructions.

Isolated peripheral T and B cells were cultured at a density of $0.5 \times 10^6$ cells/ml, in RPMI media (Gibco, 21875-034) supplemented with 10% FCS (IGC Technical Services), penicillin (70 mg/l, IGC Technical Services), streptomycin (130 mg/l, IGC Technical Services), 2 mM L-glutamine (IGC Technical Services), 1 mM sodium pyruvate (Sigma-Aldrich, S8363), 50 µM β-mercaptoethanol (Gibco, 31350-010) and 1× Non-Essential Amino Acids (Sigma-Aldrich, M7145) at 37°C. To stimulate cells, T cells were additionally cultured with 30 U/ml IL-2 (PeproTech, 212-12) and 1 µl/ml Mouse T-Activator Dynabeads (Gibco, 11,452D), whilst B cells were cultured with 10 ng/ml IL-4 (PeproTech, 214-14) and 5 µg/ml LPS (Sigma, L4391). Cells were allowed to proliferate for 48 hr prior to any auxin/BrdU treatments.

## MEFs

MEFs were derived from E13.5/E14.5 embryos. Head and organs were removed, and the embryonic body was homogenised with a sterile razor blade. 1 ml 1× trypsin (Sigma-Aldrich, T4174) in PBS was added per 3 embryos and the mixture was incubated at 37°C for 10 min. Tissue was further homogenised by passage through a 23 G needle approximately 20 times. Homogenous tissue was then resuspended in MEF media (Standard Dulbecco's Modified Eagle Medium (DMEM - Gibco, 41965-039) with 15% FCS (IGC Technical Services), penicillin (70 mg/l) and streptomycin (65 mg/l), 2 mM L-glutamine (IGC Technical Services), 1 mM sodium pyruvate (Sigma-Aldrich, S8363), 50 µM β-mercaptoethanol (Gibco, 31350-010), 1× Non-Essential Amino Acids (Sigma-Aldrich, M7145)) and passed through a 40 µm filter to remove non-homogenised tissue. MEFs were then cultured in a T75 flask (per 3 embryos) at 37°C, 5% $CO_2$, and 3% $O_2$.

## Neural stem cells

SC lines were derived from the telencephalon of individual E13.5 or E15.5 embryos following a previously described protocol (*Pollard, 2013*). Once stably propagating, NSCs were cultured in T75 flasks. When passaging, NSCs were washed with Wash Media (WM) DMEM/Ham's F-12 media with L-glutamine (Sigma-Aldrich, D8437-500), 300 mM D-(+)-glucose solution (Sigma-Aldrich, G8644), 1× MEM Non-Essential Amino Acids Solution (Gibco, 11140050), 4.5 mM HEPES (Gibco, 15630056), 75 mg/ml BSA solution, 50 µM β-mercaptoethanol (Gibco, 31350-010), penicillin (70 mg/l, IGC Technical Services), and streptomycin (130 mg/l, IGC Technical Services) and propagated in Complete Media (CM) WM supplemented with Epidermal Growth Factor (EGF) (PeproTech, 315-09) and Recombinant Human FGF-basic (FGF) (PeproTech, 100-18B) each to final concentration of 10 ng/ml, 1 µg/ml Laminin (Trevigen, 3446-005-01), 2.5 ml N-2 Supplement (100×) (Gibco, 17502048) and 5 ml B27 Supplement (50×) (Gibco, 17504044). Cells were cultured at 37°C, 5% $CO_2$ and passaged every 2–3 days.

## Auxin treatment
### Cell culture

Indole-3-acetic acid (auxin, MP Biomedicals, 102037) was solubilised in DMSO to give a 500 mM stock solution. This stock solution was then diluted in the cell media of choice to give a solution of desired concentration before being filter sterilised through a 0.22 µm filter (Starlab, E4780-1226). A DMSO-only treated sample was always processed alongside any auxin-treated sample.

## In vivo

Indole-3-acetic acid powder (125 mg) was dissolved in 1 ml PBS (Sigma-Aldrich, D8537), with small quantities of NaOH (IGC Technical Services, 5 M, 140 µl) added to help solubilise the drug. The solubilised drug was then added to 2.4 ml PBS, with minute volumes of HCl (5 M, Fisher, H/1150/PB17) added until solution pH reached 7.4. In order to achieve a more physiological osmolarity, the drug mixture was diluted to 10 ml in MQ water (final osmolarity range of ~355–380 mOsm/l; concentration = 71.4 mM). Vehicle injection mixture was prepared by adding 10 µl of both NaOH (5 M) and HCl (5 M) to 10 ml PBS (final osmolarity of 326 mOsm/l).

Both vehicle and auxin injection mixtures were filter sterilised through 0.22 µm filters prior to injection. Sterile auxin solution was then administered to animals via I.P. injection. Auxin injection volume was adjusted based on animal weight so that each animal received 100 mg of auxin per kg of body

weight. Animals were individually housed following injection, then culled using a schedule 1 approved method at appropriate timepoints, then tissues of interest were dissected and stored briefly on ice until further processing.

For in vivo auxin treatments in *Figure 6A, B*, *Figure 6—figure supplement 1B, C*, three animals per condition were injected with auxin solution and a further three animals were injected with vehicle. For in vivo auxin treatments in *Figure 6C, G*, *Figure 6—figure supplement 1A*, *Figure 6—figure supplement 2A, B, and D*, a single animal per condition was injected with auxin and a further one animal was injected with vehicle. The experimental unit was a single animal in each case. Neither a priori sample size calculation nor experimental blinding were performed. We excluded one experimental proteomics dataset (relevant to *Figure 6B*), which used whole thymus rather than sorted CD8[+] cells, due to profound contamination with red blood cells. This exclusion criterion was not pre-established. Contamination was subsequently eliminated by MACS purification for CD8[+] T-cell lineage cells (*Figure 6B*). Randomisation was not used to allocate animals to experimental groups. An ARRIVE E10 checklist for the reporting of animal experimentation has been submitted as a Supplemental data file with this manuscript.

## BrdU pulse chase assay

A 10 mM stock of BrdU was firstly generated by dissolving 0.0031 g BrdU powder (Sigma-Aldrich, B5002) in 1 ml PBS. Samples were firstly pre-depleted of Ncaph/Ncaph2 using auxin (500 µM) for 2–3 hr, before being pulsed with BrdU (final concentration = 10 µM) for 30 min at 37°C. BrdU was washed out by firstly pelleting cells via centrifugation (300 × *g*, 5 min) before washing once in media, and then pelleting samples again. Samples were then split in two, with half of the sample placed on ice for assessment of degradation efficiency at the beginning of the chase (0 hr time-point in *Figure 5—figure supplement 1C*). The other half of the sample was resuspended in pre-warmed auxin-containing media before being incubated at 37°C for a further 3.5 hr. All samples were rinsed in PBS and pelleted after their incubations were complete and stained with Fixable Viability Dye (eBioscience, 65-0865-14, 1 in 200 dilution), and/or fluorescent surface markers if required (20 min/4°C), before being washed in 2 ml Flow Cytometry staining buffer (FCSB - PBS supplemented with 2% bovine serum albumin and 2mM EDTA). To quantify the efficiency of auxin-induced depletion (*Figure 5—figure supplement 1C*), a small portion of each sample was taken at the start (0 hr) and end (3.5 hr) of the chase and analysed on the LSRFortessa. Degradation was calculated as described in 'Flow Cytometry'. Following the chase, the majority of each sample was then resuspended and fixed in Cytofix/Cytoperm solution (100 µl/sample, BD Bioscience, 554722) following manufacturer's instructions and washed in Perm/Wash buffer (2 ml/sample, BD Bioscience, 554723) before being resuspended in 0.5 ml FCSB and left overnight at 4°C.

Samples were pelleted and resuspended in Cytoperm Permeabilisation Buffer Plus (BD Bioscience, 561651) following the manufacturer's instruction, before being washed in 2 ml Perm/Wash buffer. Cytofix/Cytoperm solution (100 µl/sample) was used to re-fix samples for 5 min at 4°C before samples were again washed in 2 ml Perm/Wash buffer. DNase I solution (eBioscience, 00-4425-10 – part of BrdU Staining Kit for Flow Cytometry, 8817-6600-42) was diluted following the manufacturer's instruction. Each sample was resuspended in 100 µl diluted DNase I and incubated at 37°C for 1 hr to expose the BrdU epitope. Samples were washed in 2 ml Perm/Wash solution, before being incubated with AlexaFluor-647-conjugated anti-BrdU monoclonal antibody (Invitrogen, B35140, 1 in 20 dilution in Perm/Wash, 20-min incubation at room temperature). Perm/Wash (2 ml per sample) was used to wash samples. To stain for DNA content, each sample was resuspended in 20 µl 7-AAD (BD Biosciences, 559925) for at least 15 min before samples were finally diluted in 0.5 ml PBS. Samples were all analysed on the LSRFortessa as above.

## Immunofluorescence on tissue cryosections

Tissues were dissected immediately post-mortem then washed in ice-cold PBS, fixed for 24 hr (small intestine) or 2 hr (E10.5 embryos) in 4% PFA in PBS, then passed through 10% and 30% sucrose in PBS solutions, and mounted in OCT (Tissue-Tek). 20 µm sections were cut on a Leica CM1850 and adhered to Superfrost Plus slides (Epredia). Sections were post-fixed in 4% PFA in PBS for 10 min at room temperature, then permeablised in 0.5% Triton X-1000 for 5 min at room temperature, and washed twice in 0.2% Triton X-1000 for 5 min at room temperature. Sections were then blocked in 4% BSA

in PBS for 1 hr (small intestine) or 2 hr (E10.5 embryo) at room temperature, and primary antibodies were diluted in 4% BSA in PBS and applied overnight at 4°C. Sections were washed three times in 0.2% Triton X-1000, and secondary antibodies were diluted in 4% BSA in PBS and applied at room temperature for 2 hr. Sections were then washed as previously, stained with DAPI at 1 μg/ml, and mounted in Vectashield (Vector Labs). To reduce autofluorescence, staining of small intestine sections also included an additional treatment with 0.1% Sudan Black in 70% ethanol at room temperature for 20 min immediately prior to DAPI stain and mounting.

## Image capture and analysis

Images of small intestine sections were acquired at ×100 magnification using a Photometrics Coolsnap HQ2 CCD camera and a Zeiss AxioImager A1 fluorescence microscope with a Plan Apochromat 100 × 1.4 NA objective, a Nikon Intensilight Mercury based light source (Nikon UK Ltd, Kingston-on-Thames, UK) and either Chroma #89,014ET (3 colour) or #89,000ET (4 colour) single excitation and emission filters (Chroma Technology Corp., Rockingham, VT) with the excitation and emission filters installed in Prior motorised filter wheels. A piezoelectrically driven objective mount (PIFOC model P-721, Physik Instrumente GmbH & Co, Karlsruhe) was used to control movement in the z dimension. Hardware control and image capture were performed using Nikon Nis-Elements software (Nikon UK Ltd, Kingston-on-Thames, UK). Deconvolution of 3D data was performed in Nis-Elements (Richardson Lucy, 20 iterations). 3D datasets were visualised and analysed for fluorescence intensity using Imaris V9.5 (Bitplane, Oxford Instruments, UK). DNA volume was calculated by manually rendering a surface around the DAPI signal in pH3S10[+] cells within Imaris, and immunofluorescence signal was calculated from mean voxel intensity within that surface. The percentage protein remaining following IAA treatment was calculated by setting the mean voxel intensity measured in pH3S10[+] cells from negative control (e.g. *Ncaph*[+/+] *Ncaph2*[+/+]) sections to 0%, and the mean voxel intensity from vehicle-treated sections to 100%.

Images of embryonic whole-mount cryosections were acquired in 2D using a Zeiss Axioscan Z1 with a Plan-Apochomat 40 × 0.95 Korr M27 objective and an Axiocam 506 camera using DAPI channel as focus. Images were acquired in Zen 3.1 software, and analysed using QuPath 0.3.0 (*Bankhead et al., 2017*). Five regions per embryo were selected based on lineage marker staining (Cdh1 and Pdgfr), and mean pixel intensity was calculated in the 647 channel, corresponding to Ncaph-AID:Clover detected with an anti-GFP-647 nanobooster. The percentage protein remaining value was then calculated as described for small intestine sections.

## Materials availability statement

Proteomics data underlying *Figure 6B* ave been submitted to the PRIDE database under accession PXD032374. All other primary data, including flow cytometry files, fluorescence imaging, and uncropped western blot scans are available through the Dryad Digital Repository at https://doi.org/10.5061/dryad.g1jwstqt9.

Requests for the Rosa26[Tir1] transgenic mouse line should be addressed to Bin Gu (gubin1@msu.edu), and requests for the Ncaph- and Ncaph2-AID:Clover lines should be addressed to Andrew Wood (Andrew.j.wood@ed.ac.uk).

## Acknowledgements

We thank the University of Edinburgh BVS Central Transgenic Core for performing CRISPR microinjections to generate AID:Clover-tagged mouse lines, the Biological Research Facility at the Western General Hospital for animal husbandry, the IGC Flow Cytometry Facility, IGC Advanced Imaging Resource and IGC Mass Spectrometry facility for technical support. We acknowledge technical support from the Model Production Core staff led by M Gertsenstein at the Centre for Phenogenomics for generating the TIR1 knock-in mouse lines and Eszter Posfai (Princeton University) for early collaboration efforts on proof-of-principle studies on AID in mouse embryos. We are grateful to Luke Boulter for antibodies and advice on whole-mount immunofluorescence, to Wendy Bickmore for comments on the manuscript and to Gopal Sapkota for useful discussions. We also acknowledge Masato Kanemaki and co-workers for their pioneering efforts to develop the auxin-inducible degron system. This work was supported by a Sir Henry Dale Fellowship from the Wellcome Trust (AW 102560/Z/13/Z), an MRC Unit award to the MRC Human Genetics Unit, and Funding from CIHR (JR FDN-143334).

# Additional information

## Competing interests

Lewis Macdonald: The authors declare no competing financial interests. The other author declares that no competing interests exist.

## Funding

| Funder | Grant reference number | Author |
|---|---|---|
| Medical Research Council | Unit award to the MRC Human Genetics Unit | Lewis Macdonald<br>Gillian C Taylor<br>Jennifer Margaret Brisbane<br>Ersi Christodoulou<br>Lucy Scott<br>Andrew J Wood |
| Medical Research Council | MC_PC_21040 | Andrew J Wood<br>Gillian C Taylor |
| Wellcome Trust | 102560/Z/13/Z | Lewis Macdonald<br>Gillian C Taylor<br>Jennifer Margaret Brisbane<br>Ersi Christodoulou<br>Lucy Scott<br>Andrew J Wood |
| Canadian Institutes of Health Research | JR FDN-143334 | Janet Rossant<br>Bin Gu |

The funders had no role in study design, data collection, and interpretation, or the decision to submit the work for publication. For the purpose of Open Access, the authors have applied a CC BY public copyright license to any Author Accepted Manuscript version arising from this submission.

## Author contributions

Lewis Macdonald, Data curation, Investigation, Methodology, Validation, Visualization, Writing – review and editing; Gillian C Taylor, Conceptualization, Data curation, Formal analysis, Investigation, Supervision, Visualization, Writing – review and editing; Jennifer Margaret Brisbane, Investigation; Ersi Christodoulou, Lucy Scott, Investigation, Visualization; Alex von Kriegsheim, Data curation, Methodology, Visualization; Janet Rossant, Funding acquisition, Resources, Writing – review and editing; Bin Gu, Conceptualization, Investigation, Methodology, Resources, Writing – review and editing; Andrew J Wood, Conceptualization, Formal analysis, Funding acquisition, Investigation, Project administration, Resources, Supervision, Visualization, Writing - original draft

## Author ORCIDs

Jennifer Margaret Brisbane ⓘ http://orcid.org/0000-0003-4943-5331
Alex von Kriegsheim ⓘ http://orcid.org/0000-0002-4952-8573
Janet Rossant ⓘ http://orcid.org/0000-0002-3731-5466
Bin Gu ⓘ http://orcid.org/0000-0002-5594-2463
Andrew J Wood ⓘ http://orcid.org/0000-0002-0653-2070

## Ethics

All animal work was approved by a University of Edinburgh internal ethics committee and was performed in accordance with institutional guidelines under license by the UK Home Office. AID knock-in alleles were generated under project license PPL 60/4424. Rosa26Tir1 knock-in mouse lines were generated under the Canadian Council on Animal Care Guidelines for Use of Animals in Research and Laboratory Animal Care under protocols approved by the Centre for Phenogenomics Animal Care Committee (20-0026H). Experiments involving double transgenic animals were conducted under the authority of UK project license PPL P16EFF7EE.

## Decision letter and Author response

Decision letter https://doi.org/10.7554/eLife.77987.sa1
Author response https://doi.org/10.7554/eLife.77987.sa2

# Additional files

## Supplementary files

• Supplementary file 1. Mass spectrometry quantification of proteome-wide changes in protein level in CD8$^+$ thymocytes following IAA exposure in vivo. These data were used to generate volcano plots in *Figure 6B*.

• Supplementary file 2. Details of antibodies used in this study.

• Supplementary file 3. DNA sequences used to generate sgRNAs and donor templates for generating mouse strains via CRISPR-Cas9.

• MDAR checklist

## Data availability

Proteomics data underlying Figure 6B have been submitted to the PRIDE database under accession PXD032374. All other primary data, including flow cytometry files, fluorescence imaging and uncropped western blot scans are available through the Dryad Digital Repository at https://doi.org/10.5061/dryad.g1jwstqt9. Requests for the Rosa26Tir1 transgenic mouse line should be addressed to Bin Gu (guibin1@msu.edu), and requests for the Ncaph- and Ncaph2-AID:Clover lines should be addressed to Andrew Wood (Andrew.j.wood@ed.ac.uk).

The following datasets were generated:

| Author(s) | Year | Dataset title | Dataset URL | Database and Identifier |
|---|---|---|---|---|
| Wood AJ | 2022 | Rapid and specific degradation of endogenous proteins in mouse models using auxin-inducible degrons | https://dx.doi.org/10.5061/dryad.g1jwstqt9 | Dryad Digital Repository, 10.5061/dryad.g1jwstqt9 |
| von Kreigsheim A, Wood A | 2022 | Rapid and specific degradation of endogenous proteins in mouse models using auxin-inducible degrons | https://www.ebi.ac.uk/pride/archive/projects/PXD032374 | PRIDE, PXD032374 |

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
