## [Editor Report]

This manuscript will be of interest to the broad class of biologists and especially mouse geneticists who study the function of protein-coding genes. The authors confirm the utility of the auxin-inducible degron tool to rapidly degrade the target protein of interest by developing genetically modified mouse models. This expands the set of tools to study gene function in a cell/tissue type, in adults (bypassing embryonic lethality) and also to more finely dissect the different functions of pleiotropic genes.

---

## [Decision Letter]

**Decision letter after peer review:**

Thank you for submitting your article "Rapid and specific degradation of endogenous proteins in mouse models using auxin-inducible degrons" for consideration by *eLife*. Your article has been reviewed by 3 peer reviewers, including Guillaume Pavlovic as the Reviewing Editor and Reviewer #1, and the evaluation has been overseen by Didier Stainier as the Senior Editor. The following individuals involved in the review of your submission have agreed to reveal their identity: Lluis Montoliu (Reviewer #2); Channabasavaiah B Gurumurthy (Reviewer #3).

Essential revisions:

1) In agreement with the other two reviewers, we feel that your paper is of high quality without any major comments. This paper also does not appear to require any additional experiments.

2) A very careful reading of your paper led us to some minor comments mainly related to clarity, presentation, and improving, even more, the Discussion section. I find it interesting that as authors you incorporate these remarks, especially since they are only very minor rewrites of the article and have the potential to improve your paper even more.

*Reviewer #1 (Recommendations for the authors):*

This manuscript proposes a new technological tool that has good potential but which needs to be confirmed in the future by other studies. That is why it seems particularly interesting in this case that the tools are made easily accessible to the scientific community. The lines generated in this project, in particular, the R26-CAG-LSL-TIR1-9myc lines (before and after the cre deletion of LSL), should thus be deposited in an international repository as https://www.infrafrontier.eu/. This will greatly simplify the evaluation of this technology for other target genes.

Some of the conclusions are overly assertive (especially when it comes to addressing the drawbacks of this technology). It is only necessary to nuance them (see next).

I did not find in Appendix S1 / Table S1 the complete nucleotide sequences for the guide RNA target sequences and repair templates as indicated in the manuscript (only found the proteome data). This appendix seems missing.

This paper demonstrates the effectiveness of the system in reducing the expression of the two genes analyzed. However, insufficient arguments are developed on the possible biases of this system:

• Toxicity of IAA: what is already known and what needs still to be addressed?

• Leakiness of RosaTir1 (like a creERT vs a creERT2?) in the absence of IAA (as seen in Figure 1 G Ncaph).

• Specific phenotype from RosaTir1 line, i.e. "off-target" protein degradation by OsTir1: proteome result is a first very positive but insufficient argument: need of RosaTir1 line phenotyping and also proteome data on other organs and compared with wild-type controls (not only mutants with or without IAA). Indeed, the authors failed to generate the constitutive OsTIR1 line and hypothesize an impact of TIR1 on non-specific protein degradation (lines 149-158). The risk of TIr1-related phenotype seems therefore important.

• Mosaic expression of OsTir1: likely, Tir1 is not expressed in all cells (as no real ubiquitous line exists to date). See one of the following remarks for details.

• Impact of the integration of AID:Clover sequence in the target gene. C-terminus integration is much more likely to disturb the target gene function than, for example, in a cre system the integration of loxP in intronic sequences. In this paper Ncaph2AID:Clover homozygotes are less fertile and smaller than heterozygote by heterozygote breeding. It is therefore very likely that this line has strong phenotypes. Is the generated Ncaph2AID:Clover allele a hypomorph?

Although it seems to be unrealistic and inappropriate to ask the authors to answer these questions through experiments in this one article, it seems however indispensable to discuss more in detail these potential issues (and solutions) in the Discussion section. Especially, for the cre and creERT2 systems, the presence of the right control groups will allow to avoid misinterpretations of data. This request seems to me to be particularly important in view of the experience acquired with cre and creERT2 systems where poor (but avoidable) experimental designs have led to wrong interpretations in various publications (see this paper as an example10.1074/jbc.M512373200).

Page 3 line 65: "DNA manipulations impact protein function slowly". This is not always the case (see this paper as a contrary example 10.1038/s41598-017-08845-7). It seems more accurate to say: "DNA manipulations impact protein function through kinetics determined by".

Result: Rapid degradation of AID-tagged endogenous proteins in primary cells

• I think it would be clearer to divide this chapter into two from line 115 to 178: creation of the mouse models and then line 179-183: Rapid degradation of AID-tagged endogenous proteins in primary cells.

• In this first chapter, the phenotypic study of both Ncaph and Ncaph2 lines is far too limited to conclude that there is no damaging phenotype. In the context of this article, the fact that this has not been demonstrated does not diminish the intrinsic interest of the auxin-inducible degrons tool (proof of concept of efficiency not the absence of any drawback of the technology). I think it is however important to remain much more circumspect about the absence of phenotype in the lines and therefore to be more nuanced in the conclusion of this chapter. For example, "essential functions of NCAPH and NCAPH2 147 are largely retained by the tagged" should be nuanced, e.g. "essential functions of NCAPH and NCAPH2 147 may be largely retained by the tagged".

• "we conclude that the Rosa26osTir1 transgene did not cause biologically significant levels of auxin-independent degradation for either target protein". I'm not sure the authors can conclude this. Experiments performed here are not sufficient to demonstrate the absence of leakiness of Rosa26osTir1 transgene in the absence of auxin. The absence of leakiness of the auxin-inducible degrons tool seems to me to be essential to demonstrate, but this is the subject of another paper. Again, in my opinion, the purpose of this one is to lay the foundations for the effectiveness of the approach. Here it does not seem necessary to me to make additional experiments but rather to nuance the results.

A dedicated figure that details schematically, step by step, how this auxin-inducible degrons tool works in the mouse would be a plus for the reader.

Figure S1: section B.: M and F below Ncaph and Ncaph2 respectively: it seems it is a typo.

Figure 2: panels C, D, and E: it may be interesting to indicate the complete genotype of each cell type to facilitate understanding, e.g. Rosa26Tir1/Tir1; NcaphAID:Clover/ AID:Clover.

Figure 3 B and C. X-axis: the logarithmic scale is unreadable (characters are too small).

Figure 4 B add Rosa26Tlr1/Tlr1 genotype for greater clarity.

Figure 5 E: because Gaussian distribution cannot be assessed using a non-parametric test (Mann-Whitney) for statistical analyses.

Line 147: the data here is not sufficient to be a demonstration of retained function: "essential functions of NCAPH and NCAPH2 147 are largely retained by the tagged".

Line 314-328: some small intestine cells and some spermatocytes appeared to be resistant to degradation.

An additional hypothesis not discussed by the authors is the expression of OsTir1 under the control of the pCAG promoter from the Ros26Tir1 allele. Indeed, if we draw a parallel with so-called "ubiquitous" cre or creERT2 systems, it is clear that no promoter allows constitutive expression in all cell types (e.g. doi:10.1152/physiolgenomics.00019.2007 or Tg(CAG-Bgeo/ALPP)1Lbe/J and http://mousecre.phenomin.fr/histo/317).

Line 326: blood-testis barrier hypothesis: This barrier hypothesis preventing molecule entry has also been frequently used for tamoxifen to explain its lack of efficacy in the brain for example (blood-brain barrier).

In the brain, this hypothesis is defeated by the fact that some specific brain creERT2 deleters work well (e.g. CamK2a-creERT2) and by this paper dosing TAM and derivatives in the brain (doi: 10.3389/fncel.2016.00243). Concerning the blood-testes barrier, we can ask ourselves the same question. This paper DOI: 10.1002/mrd.22772 shows efficiency TAM mediated recombination in caput epididymis. Like tamoxifen, IAA seems to be a small molecule, an expert on the blood-brain barrier could really answer this but I personally see no arguments to prove that a small molecule would not pass the barrier.

All figures: in the body of the text, the international nomenclatures are strictly applied, but in some figures, it would be necessary to check the nomenclatures of the genes (especially italics).

Line 344: I do not believe that the current level of validation of the auxin-inducible degrons tool allows being certain that it is a widely applicable approach. I think it is more accurate to say, for example, "In this paper, we describe an approach that opens up new possibilities for studying the consequences of acute protein loss in mammalian primary cells, tissues and whole organisms."

Line 374-390: see the same point in the results, your assumptions are possible but not the only ones. Here we can rely on the knowledge of creERT2 systems to flesh out the hypotheses explaining the lack of efficiency in certain cell types: again is the OsTir1 transgene expressed or sufficiently expressed in non-responder cells?

Line 391: saturation of the system: this chapter raises the question of the use of this approach for highly expressed genes.

Line 414: "That leaky degradation did not cause problems in our study" the proteomic data are only done on CD8^+^ cells comparing mutants with or without IAA and only for Ncaph2. It is therefore far from having a global vision: this point must be nuanced.

Line 418: "If leaky degradation were a problem for other target proteins, this could be kept to a minimum through the use of Rosa26Tir1 heterozygotes". Without doing the experiment it is difficult to really give strength to this suggestion.

Again, your system is likely very close to a creERT2 approach in its biases. To properly evaluate a phenotype using auxin-inducible degrons tool, you will need at least 5 control groups (wt, wt + IAA, RosaTir, TargetAID:Clover; RosaTir + TargetAID:Clover + vehicle) in addition to the experimental group (RosaTir + TargetAID:Clover + IAA). To carry all these controls on a whole project becomes very complicated, very expensive, and requires a lot of animals (3R) but it seems also important to discuss the need for these controls.

Line 433: if the LSL cassette of your R26-CAG-LSL-TIR1-9myc line is not leaking (good transcription stop which prevents the expression of TIR protein in the presence of the LSL cassette), this line can be crossed with a well-characterized cre tissue-specific transgenic. You will end up with a triple transgenic, it is far from being simple to use and requires good controls but it is also a possibility opened by your approach.

Line 495 loxp = loxP.

Line 504 delete wild.

*Reviewer #2 (Recommendations for the authors):*

This is a beautiful sophisticated study, technically demanding but brilliantly solved, with a systematic stepwise approach, with adequate control experiments documenting every single compound and variable to be considered, which is applied to address a biological question: the role of some specific proteins present in consendin chromosomal complexes. The results obtained suggest that not all cell types and developmental stages are sensitive to the degradation of these otherwise considered essential proteins. Some cells appear to overcome the absence of these degraded proteins in order to complete their cell cycle processes. This novel finding also suggests the existence of additional compensatory mechanisms, yet to be understood, that might exist in some cells and would require further studies.

I do not have further suggestions for the authors. I consider this is a well-designed, performed, and interpreted study, prudently alerting of the potential limitations of this approach. The conclusions provided by the authors are aligned to their results and do not oversell the method, as seen in other innovative techniques that fail to be replicated. I anticipate this AID system will have a significant impact on molecular and developmental biology studies.

---

## [Author Response]

Reviewer #1 (Recommendations for the authors):This manuscript proposes a new technological tool that has good potential but which needs to be confirmed in the future by other studies. That is why it seems particularly interesting in this case that the tools are made easily accessible to the scientific community. The lines generated in this project, in particular, the R26-CAG-LSL-TIR1-9myc lines (before and after the cre deletion of LSL), should thus be deposited in an international repository as https://www.infrafrontier.eu/. This will greatly simplify the evaluation of this technology for other target genes.

These two mouse lines have been deposited to the Canadian Mouse Mutant Repository, one of the major public mouse repositories in the world. The mouse lines will be made public and distributed to researchers through CMMR following appropriate material transfer agreements upon the publication of the paper. We are currently breeding stock animals necessary to deposit the two condensin degrader lines (*Ncaph^AID:Clover;^ Rosa26^TIR1^* and *Ncaph2^AID:Clover^; Rosa26^TIR1^*) in the UK National Mouse Archive at the Mary Lyon Centre.

Some of the conclusions are overly assertive (especially when it comes to addressing the drawbacks of this technology). It is only necessary to nuance them (see next).I did not find in Appendix S1 / Table S1 the complete nucleotide sequences for the guide RNA target sequences and repair templates as indicated in the manuscript (only found the proteome data). This appendix seems missing.

We have ensured that the Appendix S1 (now named ‘Supplementary file 3’) and Table S1 (Now named ‘Supplementary file 1’) files are both present in the resubmitted manuscript.

This paper demonstrates the effectiveness of the system in reducing the expression of the two genes analyzed. However, insufficient arguments are developed on the possible biases of this system:• Toxicity of IAA: what is already known and what needs still to be addressed?• Leakiness of RosaTir1 (like a creERT vs a creERT2?) in the absence of IAA (as seen in Figure 1 G Ncaph).

We agree; the two points above are now addressed in an extended Discussion section; (line 522)

“Although the AID system presents a few important advantages over more traditional conditional genetic systems such as Cre-loxP, including qui disruption and reversibility, a few remaining issues need to be addressed after our proof-of-concept study. […] Thus, detailed knowledge of the housing and feeding condition of mouse colonies might be an important factor for the precise reproduction of experiments using AID systems.”

• Specific phenotype from RosaTir1 line, i.e. "off-target" protein degradation by OsTir1: proteome result is a first very positive but insufficient argument: need of RosaTir1 line phenotyping and also proteome data on other organs and compared with wild-type controls (not only mutants with or without IAA). Indeed, the authors failed to generate the constitutive OsTIR1 line and hypothesize an impact of TIR1 on non-specific protein degradation (lines 149-158). The risk of TIr1-related phenotype seems therefore important.

We agree and have added the following statements to address the need for more comprehensive phenotypic characterization after the section describing the mouse line generation:

Line 208:

“We conclude that the Rosa26^Tir1^ transgene did not cause levels of auxin-independent degradation that were sufficient to induce overt phenotypes in combination with either of the AID-tagged target proteins studied here. However, further comprehensive phenotyping following the guidelines and protocols established by international programs such as International Mouse Phenotyping Consortium (Birling et al., 2021), will be needed in the future to determine whether any of these genetic modifications induce phenotypic effects that were not detected in this study.”

We have also added the following sentence to the Results section immediately after the proteomic data are described (Line 364):

“However further proteomic studies are needed to assess the potential for Tir1-dependent off-target protein degradation in other cell types, and in combination with other target proteins.”

• Mosaic expression of OsTir1: likely, Tir1 is not expressed in all cells (as no real ubiquitous line exists to date). See one of the following remarks for details.

We agree and have added these sentences after the passage describing the generation of the TIR1 mouse lines (Line 169):

“However, although TIR1 expression is driven by a constitutive CAG promoter from the Rosa26 safe harbor locus, we did observe differential expression levels among the tissues tested (Figure 1: figure supplement 1E), consistent with previous observations for LacZ reporters and creERT2 (Hameyer et al., 2007; Mao, Fujiwara, and Orkin, 1999). As shown below, the expression of TIR1 in relevant tissues appears to be sufficient to facilitate target protein degradation. However, further characterizations are required to detail the expression pattern of TIR1 at tissue and single-cell levels to comprehensively characterize the TIR1 knock-in mouse lines and facilitate broader applications.”

• Impact of the integration of AID:Clover sequence in the target gene. C-terminus integration is much more likely to disturb the target gene function than, for example, in a cre system the integration of loxP in intronic sequences. In this paper Ncaph2AID:Clover homozygotes are less fertile and smaller than heterozygote by heterozygote breeding. It is therefore very likely that this line has strong phenotypes. Is the generated Ncaph2AID:Clover allele a hypomorph?Although it seems to be unrealistic and inappropriate to ask the authors to answer these questions through experiments in this one article, it seems however indispensable to discuss more in detail these potential issues (and solutions) in the Discussion section. Especially, for the cre and creERT2 systems, the presence of the right control groups will allow to avoid misinterpretations of data. This request seems to me to be particularly important in view of the experience acquired with cre and creERT2 systems where poor (but avoidable) experimental designs have led to wrong interpretations in various publications (see this paper as an example10.1074/jbc.M512373200).

We agree, and have addressed this point in the extended Discussion section, as detailed in our response to the first point (see line 524).

Page 3 line 65: "DNA manipulations impact protein function slowly". This is not always the case (see this paper as a contrary example 10.1038/s41598-017-08845-7). It seems more accurate to say: "DNA manipulations impact protein function through kinetics determined by".

We agree and have implemented this change.

Result: Rapid degradation of AID-tagged endogenous proteins in primary cells• I think it would be clearer to divide this chapter into two from line 115 to 178: creation of the mouse models and then line 179-183: Rapid degradation of AID-tagged endogenous proteins in primary cells.

We agree and have implemented this change.

• In this first chapter, the phenotypic study of both Ncaph and Ncaph2 lines is far too limited to conclude that there is no damaging phenotype. In the context of this article, the fact that this has not been demonstrated does not diminish the intrinsic interest of the auxin-inducible degrons tool (proof of concept of efficiency not the absence of any drawback of the technology). I think it is however important to remain much more circumspect about the absence of phenotype in the lines and therefore to be more nuanced in the conclusion of this chapter. For example, "essential functions of NCAPH and NCAPH2 147 are largely retained by the tagged" should be nuanced, e.g. "essential functions of NCAPH and NCAPH2 147 may be largely retained by the tagged".

We agree and have implemented this change.

• "we conclude that the Rosa26osTir1 transgene did not cause biologically significant levels of auxin-independent degradation for either target protein". I'm not sure the authors can conclude this. Experiments performed here are not sufficient to demonstrate the absence of leakiness of Rosa26osTir1 transgene in the absence of auxin. The absence of leakiness of the auxin-inducible degrons tool seems to me to be essential to demonstrate, but this is the subject of another paper. Again, in my opinion, the purpose of this one is to lay the foundations for the effectiveness of the approach. Here it does not seem necessary to me to make additional experiments but rather to nuance the results.

We agree and have modified this section to address this point as follows (Line 208):

**“**We conclude that the Rosa26^Tir1^ transgene did not cause levels of auxin-independent degradation that were sufficient to induce overt phenotypes in combination with either of the AID-tagged target proteins studied here. However, further comprehensive phenotyping following the guidelines and protocols established by international programs such as International Mouse Phenotyping Consortium (Birling et al., 2021), will be needed in the future to determine whether any of these genetic modifications induce phenotypic effects that were not detected in this study.”

We have also included an extended discussion on leaky degradation (Line 556), as detailed above.

A dedicated figure that details schematically, step by step, how this auxin-inducible degrons tool works in the mouse would be a plus for the reader.

We have included an additional Supplemental Figure (Figure 6: figure supplement 3) that details the main steps involved in setting up an AID-based approach in mice.

Figure S1: section B.: M and F below Ncaph and Ncaph2 respectively: it seems it is a typo.

This change has been implemented.

Figure 2: panels C, D, and E: it may be interesting to indicate the complete genotype of each cell type to facilitate understanding, e.g. Rosa26Tir1/Tir1; NcaphAID:Clover/ AID:Clover.

This change has been implemented.

Figure 3 B and C. X-axis: the logarithmic scale is unreadable (characters are too small).

This change has been implemented.

Figure 4 B add Rosa26Tlr1/Tlr1 genotype for greater clarity.

This change has been implemented.

Figure 5 E: because Gaussian distribution cannot be assessed using a non-parametric test (Mann-Whitney) for statistical analyses.

We found this point unclear so have been unable to address it. Figure 5E does not contain any statistical analysis. The relevant statistical test is shown in Figure 5F, which relates to the same experiment.

Line 147: the data here is not sufficient to be a demonstration of retained function: "essential functions of NCAPH and NCAPH2 147 are largely retained by the tagged".

We have addressed this point, as detailed above.

Line 314-328: some small intestine cells and some spermatocytes appeared to be resistant to degradation.An additional hypothesis not discussed by the authors is the expression of OsTir1 under the control of the pCAG promoter from the Ros26Tir1 allele. Indeed, if we draw a parallel with so-called "ubiquitous" cre or creERT2 systems, it is clear that no promoter allows constitutive expression in all cell types (e.g. doi:10.1152/physiolgenomics.00019.2007 or Tg(CAG-Bgeo/ALPP)1Lbe/J and http://mousecre.phenomin.fr/histo/317).

We have addressed this point with modified text (Line 169), as detailed above.

Line 326: blood-testis barrier hypothesis: This barrier hypothesis preventing molecule entry has also been frequently used for tamoxifen to explain its lack of efficacy in the brain for example (blood-brain barrier).In the brain, this hypothesis is defeated by the fact that some specific brain creERT2 deleters work well (e.g. CamK2a-creERT2) and by this paper dosing TAM and derivatives in the brain (doi: 10.3389/fncel.2016.00243). Concerning the blood-testes barrier, we can ask ourselves the same question. This paper DOI: 10.1002/mrd.22772 shows efficiency TAM mediated recombination in caput epididymis. Like tamoxifen, IAA seems to be a small molecule, an expert on the blood-brain barrier could really answer this but I personally see no arguments to prove that a small molecule would not pass the barrier.

The comparison with CRE-ERT2 is interesting, but Tamoxifen and IAA are different molecules (size is not the only determinant of BBB permeability) so we don’t consider this informative to discuss here.

We previously stated (Line 386) “we speculate that the blood-testes barrier could prevent IAA from entering seminiferous tubules to effect protein degradation, although cell intrinsic barriers to degradation cannot be excluded”. Given that other cell intrinsic barriers are possible beyond absence of TIR1 expression (e.g. absence of endogenous subunits of SCF^TIR1^), we consider this to be a fair and measured statement.

All figures: in the body of the text, the international nomenclatures are strictly applied, but in some figures, it would be necessary to check the nomenclatures of the genes (especially italics).

We have checked and, where necessary, amended gene nomenclature throughout the figures.

Line 344: I do not believe that the current level of validation of the auxin-inducible degrons tool allows being certain that it is a widely applicable approach. I think it is more accurate to say, for example, "In this paper, we describe an approach that opens up new possibilities for studying the consequences of acute protein loss in mammalian primary cells, tissues and whole organisms."

We have implemented the suggested change.

Line 374-390: see the same point in the results, your assumptions are possible but not the only ones. Here we can rely on the knowledge of creERT2 systems to flesh out the hypotheses explaining the lack of efficiency in certain cell types: again is the OsTir1 transgene expressed or sufficiently expressed in non-responder cells?

Figure 6: figure supplement 2D shows that the failure to degrade in Ter119+ cells is cell autonomous. As stated in line 381 of the original submission and shown in Figure 6: figure supplement 2C, Tir1 expression is no different in cells that do (CD8^+^) versus do not respond (Ter119+). To reflect the fact that we do not test TIR1 expression in the testes, we have changed the word ‘likely’ to ‘possibly’ on line 460.

Line 391: saturation of the system: this chapter raises the question of the use of this approach for highly expressed genes.

We agree and have added the underlined text to this paragraph to reflect this point (Line 478):

“Saturation of protein degradation activity is therefore a general property of molecular glue compounds, which might limit their activity on highly expressed targets and influence protein degradation efficiency in both clinical and research contexts.”

Line 414: "That leaky degradation did not cause problems in our study" the proteomic data are only done on CD8^+^ cells comparing mutants with or without IAA and only for Ncaph2. It is therefore far from having a global vision: this point must be nuanced.

The proteomic experiments in CD8^+^ cells were designed to address specificity rather than leaky degradation of the target. The relevant data are in Figure 1G and Figure 1: figure supplement 1F, G and H.

Line 418: "If leaky degradation were a problem for other target proteins, this could be kept to a minimum through the use of Rosa26Tir1 heterozygotes". Without doing the experiment it is difficult to really give strength to this suggestion.

We have removed this statement.

Again, your system is likely very close to a creERT2 approach in its biases. To properly evaluate a phenotype using auxin-inducible degrons tool, you will need at least 5 control groups (wt, wt + IAA, RosaTir, TargetAID:Clover; RosaTir + TargetAID:Clover + vehicle) in addition to the experimental group (RosaTir + TargetAID:Clover + IAA). To carry all these controls on a whole project becomes very complicated, very expensive, and requires a lot of animals (3R) but it seems also important to discuss the need for these controls.

We agree that more investigation needs to be carried out to evaluate the safety profile of the AID system, and we hope that the text additions detailed above now make this point abundantly clear in our manuscript. Our Discussion section is now very long, and while we agree with the need for the suggested controls in an ideal world, we feel that the extra points suggested here – although relevant – are better suited to a dedicated review article.

Line 433: if the LSL cassette of your R26-CAG-LSL-TIR1-9myc line is not leaking (good transcription stop which prevents the expression of TIR protein in the presence of the LSL cassette), this line can be crossed with a well-characterized cre tissue-specific transgenic. You will end up with a triple transgenic, it is far from being simple to use and requires good controls but it is also a possibility opened by your approach.

We agree and have added a statement to the discussion to explicitly state this possibility (line *519*).

“Although not tested here, the Rosa26 LOX-STOP-LOX TIR1 allele generated during this study could allow spatial restriction of IAA pharmacodynamics to cell types co-expressing Cre recombinase.”

Line 495 loxp = loxP.Line 504 delete wild.

Both of these changes have been made.